# Learn Beneficial Noise as Graph Augmentation

**Siqi Huang** [* 1 2]  **Yanchen Xu** [* 1 2]  **Hongyuan Zhang** [2 3]  **Xuelong Li** [2]

## Abstract

Although graph contrastive learning (GCL) has been widely investigated, it is still a challenge to generate effective and stable graph augmentations. Existing methods often apply heuristic augmentation like random edge dropping, which may disrupt important graph structures and result in unstable GCL performance. In this paper, we propose **P**ositive-**i**ncentive **N**oise driven **G**raph **D**ata **A**ugmentation (PiNGDA), where positive-incentive noise (pi-noise) scientifically analyzes the beneficial effect of noise under the information theory. To bridge the standard GCL and pi-noise framework, we design a Gaussian auxiliary variable to convert the loss function to information entropy. We prove that the standard GCL with pre-defined augmentations is equivalent to estimate the beneficial noise via the point estimation. Following our analysis, PiNGDA is derived from learning the beneficial noise on both topology and attributes through a trainable noise generator for graph augmentations, instead of the simple estimation. Since the generator learns how to produce beneficial perturbations on graph topology and node attributes, PiNGDA is more reliable compared with the existing methods. Extensive experimental results validate the effectiveness and stability of PiNGDA.

## 1. Introduction

With the development of Contrastive Learning (CL) in computer vision (Chen et al., 2020; He et al., 2020; Bachman et al., 2019; Falcon & Cho, 2020), natural language processing (Gao et al., 2021a; Yan et al.), and other fields(Radford et al., 2021), more and more researchers have paid great efforts on how to extend CL to graph, which is known as Graph Contrastive Learning (GCL) (Veličković et al., 2018; Zhu et al., 2021; Peng et al., 2020; Gao et al., 2021b; Rong et al., 2020). Compared with traditional contrastive learning methods, GCL considers not only how to contrast data points but also how to generate augmentation by leveraging the topological structure (Hassani & Khasahmadi, 2020; Zhu et al., 2021), local features (Veličković et al., 2018; Peng et al., 2020), and global context of graphs (Sun et al., 2020).

Despite the progress in GCL methods, graph data augmentation remains a key challenge, which is substantially different from visual data. Unlike the reliable and stable visual augmentations (*e.g.*, cropping, flipping, translation) that play the core roles in visual contrastive models (He et al., 2020; Chen et al., 2020), graph topology is more complex with non-Euclidean structure. It results in difficulties of defining efficient and stable augmentation to retain crucial topological properties.

In early GCL models, typical graph augmentations included random modifications to edges and nodes. For example, random edge dropping (Gao et al., 2021b; Rong et al., 2020) stochastically removes a portion of edges, while random node dropping (Feng et al., 2020) removes nodes and their links from the graph. Such random methods increase diversity but may disrupt inherent graph structure. More recent works propose adaptive augmentation techniques. GCA (Zhu et al., 2021) sets the probability of dropping an edge based on centrality measures of the involved nodes, aiming to better preserve graph connectivity. NCLA (Shen et al., 2023) learns adaptive graph augmentations and embeddings using a multi-head graph attention mechanism and a neighbor contrastive loss. Although providing more flexibility than completely random schemes, these predefined augmentation rules can still be considered as a priori assumptions. The heuristic methods usually lead to instability, since **the perturbations may introduce severe topological noise** and hinder GCL pre-training and downstream tasks.

To address the challenges by learning effective graph augmentations, some works have explored learning-based approaches. JOAO (You et al., 2021b) proposes a optimization framework to automatically select data augmentation methods for each sample. It treats augmentation selection as

---
[*]Equal contribution [1]School of Artificial Intelligence, OPtics and ElectroNics (iOPEN), Northwestern Polytechnical University [2]Institute of Artificial Intelligence (TeleAI), China Telecom [3]The University of Hong Kong. Correspondence to: Hongyuan Zhang <hyzhang98@gmail.com>, Xuelong Li <xuelong_li@ieee.org>.

*Proceedings of the 42$^{nd}$ International Conference on Machine Learning*, Vancouver, Canada. PMLR 267, 2025. Copyright 2025 by the author(s).

a hyperparameter optimization problem to improve GCL. Similarly, Suresh *et al.* proposes adversarial-GCL (AD-GCL) (Suresh et al., 2021b), which optimizes adversarial graph augmentation in GCL to prevent redundant information capture. It designs a practical instantiation based on trainable edge-dropping graph augmentation. These methods either select from predefined operations or focus on specific structural perturbations like edge modifications.

In summary, the existing learning-based graph augmentations introduce topological noise to graph. In this paper, we seek a *more explainable* approach and focus on how to directly control the beneficial noise with a theoretical framework, i.e., Positive-incentive Noise (Pi-Noise or $\pi$-noise) (Li, 2022) framework. $\pi$-noise is defined as the beneficial noise that reduces the task complexity. We design the **Pi-Noise** driven **G**raph **D**ata **A**ugmentation (PiNGDA). Owing to the theory about noise, the noisy graph augmentations can be easily applied to **both topology and attributes**. Note that the augmentations on node attributes are mainly heuristic methods, such as random permutation (Hassani & Khasahmadi, 2020) and random mask (Zhu et al., 2021). Roughly speaking, PiNGDA utilizes learnable noise generators to produce beneficial noise perturbations. The contributions are listed as follows:

- We design a Gaussian auxiliary variable related to the GCL training loss to quantify the GCL complexity, which bridges the $\pi$-noise framework and GCL. It shows us that the predefined augmentation is just a point estimation of $\pi$-noise, which provides a novel perspective of GCL (Section 3.3).

- The theoretical analysis directly reveals a significant drawback of the standard GCL models. The predefined augmentations that are widely used in GCL may fail to serve as strong point estimations of $\pi$-noise. In other words, they are too unreliable to be regarded as a strong priori like augmentations of vision data. We therefore propose PiNGDA to exactly minimize the $\pi$-noise principle. PiNGDA leverages a $\pi$-noise generator to learn beneficial noise for topology and attributes as augmentations (Section 3.3).

- To ensure the differentiability of noise generation, we develop an efficient differentiable algorithm (Section 4).

- By learning the noise generative model, PiNGDA improves GCL performance and stability compared to baselines from the extensive experiments (Section 5).

## 2. Related Work

### 2.1. Graph Contrastive Learning

Recent advancements in graph contrastive learning methods (Grover & Leskovec, 2016; Hamilton et al., 2017b; Perozzi et al., 2014; Hassani & Khasahmadi, 2020) have introduced a variety of data augmentation strategies, such as node dropping (You et al., 2020), edge perturbation (Zhu et al., 2020), and feature masking (Zhu et al., 2021). While effective in creating diverse graph views, these traditional techniques struggle with the inherent complexity of graph topology. Recognizing these limitations, more adaptive approaches have been developed to better reflect the intrinsic properties of graph data. One notable example is GCA (Zhu et al., 2021), which proposes an augmentation method that adapts both topological and semantic aspects of the graph. Recent innovations such as JOAO (You et al., 2021b) and AD-GCL (Suresh et al., 2021b) have been developed to address these challenges through more sophisticated strategies. JOAO employs an augmentation-aware projection head that dynamically selects augmentations during training. AD-GCL adopts an adversarial approach where the model learns to optimize edge-dropping strategies in real-time, allowing it to adaptively modify the graph structure to enhance generalization and robustness in learned representations.

### 2.2. Methods of Learning Graph Structure

Except for GCL, graph structure learning aims at modifying the structure of a graph to achieve specific goals (Hu et al., 2019; Veličković et al., 2017; Kipf & Welling, 2016; Xu et al., 2018; Wu et al., 2019; Hamilton et al., 2017a). While these methods differ from the heuristic approach of edge dropping, they are still an important research direction. For example, a neural network-based method (Zheng et al., 2020) automatically selects the most important nodes and edges to obtain a more compact and robust graph representation. VIB-GSL (Sun et al., 2022) employs variational inference to learn the structure of a graph by maximizing the information bottleneck of its representation. In summary, these methods typically require supervised information when modifying the graph structure, which differs from the unsupervised nature of GCL.

### 2.3. The Positive Impact of Noise

Noise may not always be harmful as it has been demonstrated that random noise helps improving performance, *e.g.*, random forest (Breiman, 2001), Dropout (Srivastava et al., 2014), noisy augmentations in contrastive learning (Verma et al., 2021; Chen et al., 2020). Verma *et al.* (Verma et al., 2021) proposed a method that introduces noise in the form of diverse transformations to improve contrastive learning across different domains. The paper (Lin et al.,

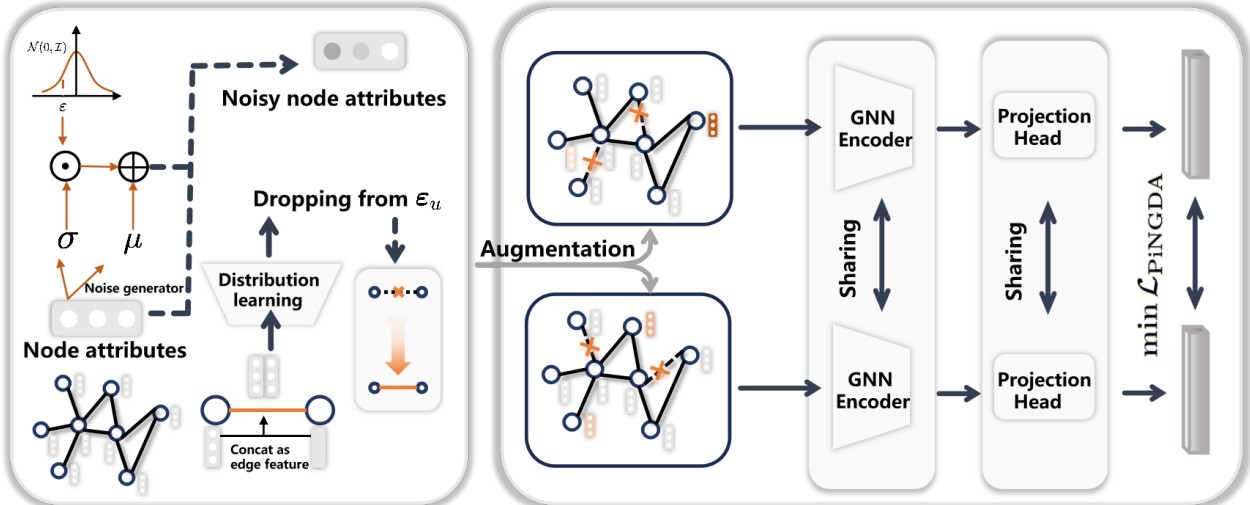

Figure 1: This figure illustrates the PiNGDA framework, which consists of a $\pi$-noise generator and contrastive learning module. The core innovation lies in its joint noise generator, which consists of two synergistic components: a topological generator for perturbing graph structures and an attribute generator for perturbing node features. A contrastive loss function is applied between original and noise graph representations. By jointly training the generator and encoder, PiNGDA dynamically learns optimal perturbation patterns tailored to downstream tasks.

2023) proposes a CNN-based hydroacoustic signal recognition method for ship noise classification. VPN (Zhang et al., 2023) shows that certain types of noise can benefit model. Similarly, other methods (Zhang et al., 2024; Huang et al., 2025) extended this insight to contrastive learning and vision-language alignment, respectively. EPAGCL (Xu et al., 2025) demonstrates that introducing structurally effective noise into graph structures can enhance representation quality. In this paper, we follow the definition in (Li, 2022). Formally, the noise $\varepsilon \in \mathcal{E}$, where $\mathcal{E}$ represents the noise set is defined as the $\pi$-noise if it satisfies

$$I(\mathcal{T}, \mathcal{E}) > 0 \Leftrightarrow H(\mathcal{T}) > H(\mathcal{T}|\mathcal{E}). \qquad (1)$$

$H(\mathcal{T})$ represents the information entropy of task $\mathcal{T}$ and $I(\cdot, \cdot)$ denotes the mutual information. It informs us that the additional components should reduce task uncertainty $H(\mathcal{T})$ after introducing noise. From this perspective, we can find that the predefined augmentations, widely used in existing GCL, may not reduce the task entropy. In this paper, we start from the above framework to learn the noisy graph augmentation with theoretical guarantee.

## 3. $\pi$-Noise Driven Graph Data Augmentation

In this section, we focus on how to apply $\pi$-Noise to GCL. As mentioned previously, it is imperative to mathematically quantify the difficulty of the target task $\mathcal{T}$. Roughly speaking, we design an auxiliary Gaussian variable based on the GCL training loss and compute its information entropy to quantify the complexity of task $\mathcal{T}$. We assume that *the*

*contrastive element is the node* for simplicity, while **it is straightforward to extend the analysis to graph-level GCL models as well as hybrid models, which uses both graphs and nodes for contrast**. Although we only discuss the node-level GCL in Section 3 and 4, **we also report the experimental results of graph tasks in Section 5**. In the succeeding sections, a graph $\mathcal{G} = (\mathcal{V}, \mathcal{S})$ consists of node set $\mathcal{V}$ and edge set $\mathcal{S}$. $\boldsymbol{\theta}^*$ denotes the optimal solution minimizing the loss $\mathcal{L}(\mathcal{V}; \boldsymbol{\theta})$ where $\boldsymbol{\theta}$ represents parameters.

### 3.1. General Formulation of GCL

Before the formal analysis, on a given graph $\mathcal{G} = (\mathcal{V}, \mathcal{S})$, we formulate the base GCL loss based on InfoNCE (van den Oord et al., 2018) as

$$\mathcal{L}_{\text{InfoNCE}} = -\frac{1}{|\mathcal{V}|} \sum_{\boldsymbol{u} \in \mathcal{V}} \log \frac{\ell_{\text{pos}}(\boldsymbol{u}; \boldsymbol{\theta})}{\ell_{\text{pos}}(\boldsymbol{u}; \boldsymbol{\theta}) + \ell_{\text{neg}}(\boldsymbol{u}; \boldsymbol{\theta})}. \quad (2)$$

$\ell_{\text{pos}}(\boldsymbol{u}; \boldsymbol{\theta})$ and $\ell_{\text{neg}}(\boldsymbol{u}; \boldsymbol{\theta})$ represent the positive and negative loss associated with node $\boldsymbol{u}$ parameterized by $\boldsymbol{\theta}$, respectively. $|\mathcal{V}|$ is the node number. *Note that there is usually a temperature hyper-parameter $\tau$ in $\ell_{\text{pos}}$ and $\ell_{\text{neg}}$. Since we aim at learning $\pi$-Noise for a specific GCL model (i.e., fixed $\tau$), we assume that a GCL model is given with a fixed $\tau$ and the hyperparameter $\tau$ is therefore omitted.*

To simplify the analysis, the contrastive loss can be separated into multiple node-level formulations

$$\ell(\boldsymbol{u}; \boldsymbol{\theta}) = -\log \frac{\ell_{\text{pos}}(\boldsymbol{u}; \boldsymbol{\theta})}{\ell_{\text{pos}}(\boldsymbol{u}; \boldsymbol{\theta}) + \ell_{\text{neg}}(\boldsymbol{u}; \boldsymbol{\theta})}, \qquad (3)$$

so that $\mathcal{L}_{\text{InfoNCE}} = 1/|\mathcal{V}| \cdot \sum_{\boldsymbol{u}} \ell(\boldsymbol{u}; \boldsymbol{\theta})$.

## 3.2. A General Framework to Bridge $\pi$-Noise and Training Loss

Given an arbitrary loss function $\mathcal{L}(\mathcal{V}; \boldsymbol{\theta}) = \sum_{\boldsymbol{u} \in \mathcal{V}} \ell(\boldsymbol{u}; \boldsymbol{\theta})$, let $\boldsymbol{\theta}^*$ be the optimum that minimizes $\mathcal{L}(\mathcal{V}; \boldsymbol{\theta})$ over the parameter space. It is clear that the quantity $\mathcal{L}(\mathcal{V}; \boldsymbol{\theta}^*)$ usually provides a direct measurement of the task difficulty. In other words, a smaller $\mathcal{L}(\mathcal{V}; \boldsymbol{\theta}^*)$ implies a simpler task on $\mathcal{V}$. However, it is important to point out that $\mathcal{L}(\mathcal{V}; \boldsymbol{\theta}^*)$ is neither a random variable nor a probability quantity, so it is impracticable to be directly applied to computing the task entropy. To address this problem, we introduce **an auxiliary random variable** $\alpha$. To keep notations uncluttered, we simply substitute $\boldsymbol{\theta}^*$ with $\boldsymbol{\theta}$. We define a monotonously increasing mapping function $f : \mathbb{R} \mapsto \mathbb{R}_+$ such that

$$p(\alpha|\boldsymbol{u}) = \mathcal{N}\Big(0, f\big(\ell(\boldsymbol{u}; \boldsymbol{\theta})\big)\Big). \tag{4}$$

Here $\mathcal{N}(\cdot, \cdot)$ represents a Gaussian distribution. Due to the monotonous increasing property of $f$, **a smaller contrastive loss implies a higher similarity between samples and corresponds to a smaller variance of its auxiliary Gaussian distribution.** It indicates a smaller entropy $H\big(p(\alpha|\boldsymbol{u})\big)$, *i.e.*, a much easier task. The above definition provides us a natural scheme to define the task entropy of a given task by converting its loss $\ell(\boldsymbol{u}; \boldsymbol{\theta})$. Formally,

$$\begin{aligned} H(\mathcal{T}) &= \mathbb{E}_{\boldsymbol{u} \sim p(\boldsymbol{u})} H\big(p(\alpha|\boldsymbol{u})\big) \\ &= \mathbb{E}_{\boldsymbol{u} \sim p(\boldsymbol{u})} H\Big(\mathcal{N}\big(0, f(\ell(\boldsymbol{u}; \boldsymbol{\theta}))\big)\Big). \end{aligned} \tag{5}$$

## 3.3. Rethink GCL under $\pi$-Noise Framework

With the framework formulated in the previous subsection, we provide a new perspective for the existing GCL models. We show that a general GCL model with predefined augmentations is equivalent to learning parameters with a point estimation of $\pi$-noise. Under the $\pi$-noise framework, as the optimal parameters $\boldsymbol{\theta}^*$ are unknown, the optimization principle of GCL should be

$$\max_{\mathcal{E}, \boldsymbol{\theta}} -H(\mathcal{T}|\mathcal{E}). \tag{6}$$

According to the definition of $H(\mathcal{T})$ shown in Eq. (5), the conditional entropy is formulated as

$$\begin{aligned} &H(\mathcal{T}|\mathcal{E}) \\ &= -\int p(\alpha|\boldsymbol{u}, \boldsymbol{\varepsilon}) p(\boldsymbol{\varepsilon}|\boldsymbol{u}) p(\boldsymbol{u}) \log p(\alpha|\boldsymbol{u}, \boldsymbol{\varepsilon}) d\boldsymbol{u} d\boldsymbol{\varepsilon} d\alpha \\ &\approx -\frac{1}{n} \sum_{\boldsymbol{u}} \int p(\alpha|\boldsymbol{u}, \boldsymbol{\varepsilon}) p(\boldsymbol{\varepsilon}|\boldsymbol{u}) \log p(\alpha|\boldsymbol{u}, \boldsymbol{\varepsilon}) d\boldsymbol{\varepsilon} d\alpha. \end{aligned} \tag{7}$$

In this step, we use Monte Carlo method since the dataset $\mathcal{V}$ can be regarded as a sample drawn from $p(\boldsymbol{u})$. For GCL,

we let $f(\cdot) = \exp(\cdot)$ so that

$$p(\alpha|\boldsymbol{u}) = \mathcal{N}(0, \kappa_{\boldsymbol{\theta}}(\boldsymbol{u})^{-1}), \tag{8}$$

where $\kappa_{\boldsymbol{\theta}}(\boldsymbol{u}) = \frac{\ell_{\text{pos}}(\boldsymbol{u};\boldsymbol{\theta})}{\ell_{\text{pos}}(\boldsymbol{u};\boldsymbol{\theta}) + \ell_{\text{neg}}(\boldsymbol{u};\boldsymbol{\theta})}$ is defined to keep the derivation uncluttered.

There are two probabilities remained in Eq. (7) to be discussed. $p(\boldsymbol{\varepsilon}|\boldsymbol{u})$ is the distribution of $\pi$-noise we want to learn. A critical step in the process is to model $p(\alpha|\boldsymbol{u}, \boldsymbol{\varepsilon})$ accurately. We define $\ell_{\text{pos}}(\boldsymbol{u}, \boldsymbol{\varepsilon}; \boldsymbol{\theta})$ and $\ell_{\text{neg}}(\boldsymbol{u}, \boldsymbol{\varepsilon}; \boldsymbol{\theta})$ as contrastive losses with augmentation views generated by $\boldsymbol{\varepsilon}$. Then we follow the definitions of Eq. (4) and Eq. (8) to define

$$p(\alpha|\boldsymbol{u}, \boldsymbol{\varepsilon}) = \mathcal{N}(0, \kappa_{\boldsymbol{\theta}}(\boldsymbol{u}, \boldsymbol{\varepsilon})^{-1}), \tag{9}$$

where $\kappa_{\boldsymbol{\theta}}(\boldsymbol{u}, \boldsymbol{\varepsilon}) = \frac{\ell_{\text{pos}}(\boldsymbol{u}, \boldsymbol{\varepsilon}; \boldsymbol{\theta})}{\ell_{\text{pos}}(\boldsymbol{u}, \boldsymbol{\varepsilon}; \boldsymbol{\theta}) + \ell_{\text{neg}}(\boldsymbol{u}, \boldsymbol{\varepsilon}; \boldsymbol{\theta})}$. Without loss of generality, we *suppose that only one augmentation is used for contrast*. Clearly, the following analysis can be easily extended to the case with multiple augmentations. If the given data augmentation is regarded as a **strongly definite noise**, then augmentation can be viewed as a hypothesis as follows

$$p(\boldsymbol{\varepsilon}|\boldsymbol{u}) \to \delta_{\boldsymbol{\varepsilon}_0}(\boldsymbol{\varepsilon}), \tag{10}$$

where $\delta_{\boldsymbol{\varepsilon}_0}(\boldsymbol{\varepsilon})$ is the Dirac delta function with translation $\boldsymbol{\varepsilon}_0$, *i.e.*, $\delta_{\boldsymbol{\varepsilon}_0}(\boldsymbol{\varepsilon}) = 0$ except for $\boldsymbol{\varepsilon} = \boldsymbol{\varepsilon}_0$ and $\int_{\varepsilon} \delta_{\boldsymbol{\varepsilon}_0}(\boldsymbol{\varepsilon}) d\boldsymbol{\varepsilon} = 1$. With this assumption, $-H(\mathcal{T}|\mathcal{E})$ is equivalent to

$$-H(\mathcal{T}|\mathcal{E}) \approx \frac{1}{n} \sum_{\boldsymbol{u}} \int p(\alpha|\boldsymbol{u}, \boldsymbol{\varepsilon}_0) \log p(\alpha|\boldsymbol{u}, \boldsymbol{\varepsilon}_0) d\alpha = \mathcal{L}. \tag{11}$$

We can expand the density of $\mathcal{N}(0, \kappa_{\boldsymbol{\theta}}(\boldsymbol{u}, \boldsymbol{\varepsilon})^{-1})$ and substitute it into $\mathcal{L}$,

$$\mathcal{L} = \frac{1}{n} \sum_{\boldsymbol{u}} \Big( \log C + \frac{1}{2} \log \kappa_{\boldsymbol{\theta}}(\boldsymbol{u}, \boldsymbol{\varepsilon}_0) - \frac{1}{2} \Big), \tag{12}$$

where $C$ in the above equation represents a constant independent of learnable parameters. The detailed derivations can be found in Appendix A. To sum up, the original goal to maximize the mutual information is converted to

$$\begin{aligned} &\max_{\mathcal{E}, \boldsymbol{\theta}} I(\mathcal{T}, \mathcal{E}) \Leftrightarrow \max_{\boldsymbol{\theta}} \frac{1}{n} \sum_{\boldsymbol{u}} \log \kappa_{\boldsymbol{\theta}}(\boldsymbol{u}, \boldsymbol{\varepsilon}_0) \\ &\Leftrightarrow \min \frac{1}{n} \sum_{\boldsymbol{u}} -\log \frac{\ell_{\text{pos}}(\boldsymbol{u}, \boldsymbol{\varepsilon}_0; \boldsymbol{\theta})}{\ell_{\text{pos}}(\boldsymbol{u}, \boldsymbol{\varepsilon}_0; \boldsymbol{\theta}) + \ell_{\text{neg}}(\boldsymbol{u}, \boldsymbol{\varepsilon}_0; \boldsymbol{\theta})}, \end{aligned} \tag{13}$$

which is the same as the standard contrastive learning paradigm.

In conclusion, the standard contrastive learning paradigm is equivalent to **optimizing a contrastive learning module with a point estimation of the $\pi$-noise**, where the predefined data augmentation is the point estimation. It is easy to obtain a further conclusion: the heuristic graph augmentation **often fails to be a good point-estimation of $\pi$-noise** (shown in Eq. (10)), which causes the instability of GCL using heuristic graph augmentations.

## 3.4. Loss of $\pi$-Noise Driven Data Augmentation

Driven by the analysis of the previous subsection, there is a natural idea to obtain a stable augmentation for GCL: *learning* $\pi$-noise by optimizing Eq. (6), instead of randomly editing edges/nodes in a heuristic way (Gao et al., 2021b; Rong et al., 2020; Feng et al., 2020). It is named **P**i-**N**oise driven **G**raph **D**ata **A**ugmentation (PiNGDA). It should be emphasized that PiNGDA is *fully compatible* with existing graph contrastive learning models.

Compared with the visual contrastive learning, there is no stable method to generate an augmented graph with reliable topology and attributes. The instable augmentation implies that Eq. (10) fails to hold with a high probability. It results in the biased training of GCL models. In our proposed method, we denote $p_\psi(\varepsilon|\boldsymbol{u})$ as the learnable distribution with $\pi$-noise generator function parameterized by $\psi$ which will be discussed in Section 4. Then $H(\mathcal{T}|\mathcal{E})$ estimated by the Monte Carlo method (formulated by Eq. (7)) can be rewritten as $H(\mathcal{T}|\mathcal{E}) \approx$

$$\frac{1}{n} \sum_{\boldsymbol{u}} \int \mathbb{E}_{\varepsilon \sim p_\psi(\varepsilon|\boldsymbol{u})} p(\alpha|\boldsymbol{u}, \varepsilon) \log p(\alpha|\boldsymbol{u}, \varepsilon) d\alpha. \quad (14)$$

Accordingly, the loss of PiNGDA is formulated as

$$\mathcal{L}_\pi = -\frac{1}{n} \sum_{\boldsymbol{u}} \mathbb{E}_{\varepsilon \sim p_\psi(\varepsilon|\boldsymbol{u})} \log \frac{\ell_{\text{pos}}(\boldsymbol{u}, \varepsilon; \boldsymbol{\theta})}{\ell_{\text{pos}}(\boldsymbol{u}, \varepsilon; \boldsymbol{\theta}) + \ell_{\text{neg}}(\boldsymbol{u}, \varepsilon; \boldsymbol{\theta})}, \quad (15)$$

where $\boldsymbol{\theta}$ represents the parameters of the contrast model. For node $\boldsymbol{u}$, the feature embedding $\boldsymbol{z}_{\boldsymbol{u}}$ in the original graph and the augmented $\boldsymbol{z}_{\boldsymbol{u}}^\varepsilon$ generated $\pi$-noise constitutes a positive sample. And the other nodes are treated as negative samples. Specifically speaking, the formulations of $\ell_{\text{pos}}$ and $\ell_{\text{neg}}$ are

$$\begin{cases} \ell_{\text{pos}}(\boldsymbol{u}) = \exp\left(\frac{\langle \boldsymbol{z}_{\boldsymbol{u}}, \boldsymbol{z}_{\boldsymbol{u}}^\varepsilon \rangle}{\tau}\right), \\ \ell_{\text{neg}}(\boldsymbol{u}) = \exp\left(\frac{\langle \boldsymbol{z}_{\boldsymbol{u}}, \boldsymbol{z}_{\boldsymbol{v}} \rangle}{\tau}\right) + \exp\left(\frac{\langle \boldsymbol{z}_{\boldsymbol{u}}, \boldsymbol{z}_{\boldsymbol{v}}^\varepsilon \rangle}{\tau}\right), \end{cases} \quad (16)$$

where $\langle \cdot, \cdot \rangle$ is the similarity function and the cosine similarity is used in this paper.

Since a graph neural network usually takes two components as input, edge set and node attributes, the noise also consists of two components, topological noise $\varepsilon^{\text{edge}}$ and attribute noise $\varepsilon^{\text{attr}}$. In Section 4, we will discuss how to generate the topological noise and attribute noise, respectively.

# 4. Implementation Details of Topological Noise and Attribute Noise

In this section, we provide a detailed explanation of the implementation of both topological and attribute noise generation within our framework. We first discuss the generation of topological noise $\varepsilon^{\text{edge}}$, and then describe how

---

**Algorithm 1** Pseudo code of PiNGDA

**Input:** $\mathcal{G} = (\mathcal{V}, \mathcal{S})$: Graph data with nodes $\mathcal{V}$ and edges $\mathcal{S}$, $k$: number of loops
**Output:** Node embeddings of $\mathcal{G}$
**for** epoch in range(max_epochs) **do**
  Generate topological noise $\varepsilon^{\text{edge}}$
  Generate attribute noise $\varepsilon^{\text{attr}}$
  Get the noisy graph view $\mathcal{G}_\varepsilon$.
  Obtain node embeddings $\boldsymbol{Z} = \text{Enc}(\mathcal{G}), \boldsymbol{Z}^\varepsilon = \text{Enc}(\mathcal{G}_\varepsilon)$.
  Compute the loss $\mathcal{L}_\pi$ by Eq. (15)
  Update parameters by applying stochastic gradient ascent to minimize $\mathcal{L}_\pi$ by Eq. (15) and Eq. (16)
**end for**

---

node attribute noise $\varepsilon^{\text{attr}}$ is generated using a parameterized Gaussian distribution . The whole algorithm is summarized in Algorithm 1.

### 4.1. Topological Noise

In this subsection, our goal is to model the conditional distribution of topological noise $p(\varepsilon^{\text{edge}}|\boldsymbol{u})$, which determines edge-dropping probabilities for node $\boldsymbol{u}$. To keep simplicity, we assume the topological noise $\varepsilon^{\text{edge}}$ follows a factored distribution over edges connected to node $\boldsymbol{u}$,

$$p(\varepsilon^{\text{edge}}|\boldsymbol{u}) = \prod_{\langle \boldsymbol{u}, \boldsymbol{v} \rangle \in \mathcal{S}} p(\varepsilon_{\boldsymbol{v}}|\boldsymbol{u}), \quad (17)$$

where $\mathcal{S}$ denotes the set of edges incident to node $\boldsymbol{u}$. Each edge $\langle \boldsymbol{u}, \boldsymbol{v} \rangle$ may be dropped subject to

$$\Pr(\text{dropping } \langle \boldsymbol{u}, \boldsymbol{v} \rangle) = p(\varepsilon_{\boldsymbol{v}}|\boldsymbol{u}). \quad (18)$$

The edge-dropping probability is parameterized by a learnable module $g_\psi(\boldsymbol{u}, \boldsymbol{v})$ that operates on node pairs. While different distribution families can be employed, we use the Bernoulli distribution as our default choice due to its natural interpretation for binary edge operations. For each edge, the preceding probability is formulated as

$$p(\varepsilon_{\boldsymbol{v}}|\boldsymbol{u}) = \text{Bernoulli}(g_\psi(\boldsymbol{u}, \boldsymbol{v})) \quad (19)$$

where $g_\psi(\boldsymbol{u}, \boldsymbol{v})$ outputs the probability of edge deletion. We implement $g_\psi$ as a two-layer Multi-Layer Perceptron (MLP) that processes concatenated node features.

To maintain gradient flow through discrete edge-dropping decisions, we employ the Gumbel-Softmax reparameterization trick (Jang et al., 2016). This provides a differentiable approximation of Bernoulli sampling:

$$\varepsilon_{\boldsymbol{v}} = \text{Gumbel-Softmax}(g_\psi(\boldsymbol{u}, \boldsymbol{v})) \quad (20)$$

This formulation enables learnable, node-specific edge dropout patterns that adapt during training while maintaining differentiability through the Gumbel-Softmax approximation.

### 4.2. Attribute Noise

In this subsection, we detail the process of generating attribute noise $\varepsilon^{\text{attr}}$. The objective is to introduce noise to the node attributes in a way that is both effective for model training and differentiable. We model the attribute noise $\varepsilon^{\text{attr}}$ as a Gaussian distribution conditioned on the node attributes $\boldsymbol{u}$,

$$p(\varepsilon^{\text{attr}}|\boldsymbol{u}) = \mathcal{N}(\boldsymbol{\mu}, \boldsymbol{\Sigma}), \tag{21}$$

where $\boldsymbol{\mu}$ is the learnable mean vector and $\boldsymbol{\Sigma}$ is the learnable covariance matrix of the distribution. These parameters are learned by an MLP that takes the node attributes $\boldsymbol{u}$ as input. Similarly, we employ the reparameterization trick (Kingma, 2013) to ensure that the model is differentiable. Similar to (Kingma, 2013), we assume the Gaussian noise on attribute, which is discussed in Section 4.2, is uncorrelated, *i.e.*, $\boldsymbol{\Sigma} = \text{diag}(\boldsymbol{\sigma}^2)$ where $\boldsymbol{\sigma} \in \mathbb{R}^d$. The assumption can drastically decrease the parameters from $d^2$ to $d$. Using the reparameterization trick, the attribute noise can be generated by

$$\varepsilon^{\text{attr}} = \boldsymbol{\mu} + \boldsymbol{\sigma} \cdot \boldsymbol{\epsilon}, \tag{22}$$

where $\boldsymbol{\epsilon} \sim \mathcal{N}(0, \boldsymbol{I})$ is a standard normal random variable. This reparameterization ensures that the sampling process is differentiable, enabling backpropagation of gradients through the noise term during training.

We then add the noise to the attributes to obtain the perturbed attributes. By introducing noise into the node attributes in this way, we effectively perturb the node features.

## 5. Experiments

In this section, we conduct experiments to evaluate our model by answering the following questions.

• **Q1**: Does our proposed PiNGDA on graph outperform existing baseline methods?

• **Q2**: Is the $\pi$-noise we trained more useful than random noise? How does each component affect model performance?

• **Q3**: How does our method perform in terms of time and space efficiency?

• **Q4**: How does the $\pi$-noise look like?

We begin with a brief introduction of experimental settings, followed by a detailed presentation of the experimental results and their analysis.

### 5.1. Experimental Settings

For every experiment of node classification, we follow the linear evaluation scheme introduced by Veličković et al. (Veličković et al., 2018). Firstly, every model is trained in an unsupervised manner and then the resulting embeddings are utilized to train and test a simple $l_2$-regularized logistic regression classifier. For each dataset, we conducted 20 random splits of training /validation/test, and reported the averaged performance. We then calculated the average performance on each dataset based on these runs. In these experiments, we measured performance using accuracy as the evaluation metric. For graph classfication, all methods are trained with the corresponding self-supervised objective and then evaluated with a linear classifier. We follow the conventional 10-Fold evaluation. All our experiments are performed 10 times with different random seeds and we report mean and standard deviation of the corresponding test metric for each dataset. We use accuracy to measure the performance.

### 5.2. Baselines

For node classification, we compare PiNGDA with state-of-the-art methods for node classification. These methods include two supervised graph neural networks (GNNs), namely GCN (Kipf & Welling, 2016) and GAT (Veličković et al., 2017). Additionally, we compare PiNGDA with self-supervised GCL methods, which are DGI (Veličković et al., 2018), GMI (Peng et al., 2020), GCA (Zhu et al., 2021), BGRL (Thakoor et al., 2021), GREET (Liu et al., 2023), SGRL (He et al., 2024), GRACEIS (Liu et al., 2024), GOUDA (Zhuo et al., 2024), GASSL (Yang et al., 2021) and two learnable method AD-GCL (Suresh et al., 2021a) and JOAO (You et al., 2021a). These methods represent the current state-of-the-art in the field of semi-supervised node classification, and we compare the performance of PiNGDA against them to evaluate its effectiveness. For all baselines, we implement them based on their official codes and conduct a hyperparameter search according to the original paper.

### 5.3. Performance on Graph Tasks (Q1)

#### 5.3.1. Node Classification Results

In Table 1, we present the node classification accuracy results of various methods across seven benchmark datasets. The bold numbers indicate the best performance and underlined numbers represent the second-highest accuracy for each dataset. We applied two types of augmentation to our method: learning the beneficial noise on both the topology and attributes through a trainable noise generator. A detailed analysis of these components will be provided in the following ablation study.

Table 1: Node classification accuracy on seven datasets. The bold numbers represent the best results and the second highest results are underlined. OOM indicates the Out-Of-Memory exception on a 24GB GPU. *JOAO is marked with an asterisk to clarify that it does not learn augmentation parameters directly. It adaptively selects augmentations based on a similarity-driven policy.

| Methods | Learnable | Cora | CiteSeer | PubMed | Wiki-CS | Amazon-Photo | Coauthor-Phy | ogbn-arxiv | Rank |
|---------|-----------|------|----------|--------|---------|--------------|--------------|------------|------|
| GCN | ✗ | $84.22 \pm 1.14$ | $71.59 \pm 0.75$ | $85.19 \pm 0.21$ | $80.27 \pm 0.55$ | $92.55 \pm 0.25$ | $95.75 \pm 0.05$ | $\mathbf{69.42 \pm 0.06}$ | - |
| GAT | ✗ | $83.74 \pm 0.50$ | $72.11 \pm 0.36$ | $84.79 \pm 0.28$ | $80.41 \pm 0.58$ | $92.48 \pm 0.21$ | $95.40 \pm 0.08$ | OOM | - |
| DGI | ✗ | $83.25 \pm 0.77$ | $72.01 \pm 0.83$ | $85.12 \pm 0.18$ | $79.34 \pm 0.42$ | $89.01 \pm 0.98$ | OOM | OOM | 7.2 |
| GMI | ✗ | $83.60 \pm 1.06$ | $69.11 \pm 1.07$ | $84.12 \pm 0.37$ | $79.99 \pm 0.51$ | $90.13 \pm 1.39$ | OOM | OOM | 8 |
| GCA | ✗ | $\underline{85.20 \pm 0.21}$ | $71.76 \pm 0.24$ | $\underline{87.07 \pm 0.25}$ | $81.26 \pm 0.11$ | $93.19 \pm 0.24$ | $94.96 \pm 0.25$ | $\underline{69.23 \pm 0.01}$ | 3.4 |
| BGRL | ✗ | $83.80 \pm 0.68$ | $71.51 \pm 0.65$ | $85.60 \pm 0.17$ | $\underline{81.38 \pm 0.14}$ | $93.19 \pm 0.43$ | $95.54 \pm 0.06$ | $68.60 \pm 0.23$ | 4.2 |
| GREET | ✗ | $80.47 \pm 0.55$ | $\underline{72.28 \pm 0.59}$ | $86.07 \pm 0.53$ | $79.92 \pm 0.28$ | $\mathbf{93.61 \pm 0.35}$ | $\mathbf{96.05 \pm 0.12}$ | OOM | 4.3 |
| SGRL | ✗ | $83.63 \pm 0.49$ | $71.89 \pm 0.25$ | $86.34 \pm 0.16$ | $81.28 \pm 0.16$ | $92.97 \pm 0.32$ | $95.64 \pm 0.10$ | $68.03 \pm 0.04$ | 4.3 |
| GRACEIS | ✗ | $84.57 \pm 0.41$ | $71.40 \pm 0.53$ | $84.55 \pm 0.37$ | $79.33 \pm 0.27$ | $91.26 \pm 0.58$ | $94.28 \pm 0.43$ | $65.97 \pm 0.20$ | 7.1 |
| GOUDA | ✗ | $82.25 \pm 0.54$ | $70.25 \pm 1.24$ | $85.82 \pm 0.71$ | - | $89.61 \pm 1.17$ | $94.54 \pm 0.59$ | - | - |
| GASSL | ✗ | $81.82 \pm 0.79$ | $69.51 \pm 0.84$ | $84.91 \pm 0.57$ | - | $92.14 \pm 0.23$ | $94.93 \pm 0.21$ | - | - |
| AD-GCL | ✓ | $83.88 \pm 0.25$ | $68.74 \pm 0.53$ | $84.23 \pm 0.21$ | $80.57 \pm 0.27$ | $91.92 \pm 0.37$ | $95.63 \pm 0.10$ | $68.45 \pm 0.11$ | 6 |
| JOAO | ✓* | $82.77 \pm 0.71$ | $72.09 \pm 0.17$ | $83.83 \pm 0.32$ | $80.43 \pm 0.27$ | $78.23 \pm 2.63$ | $94.30 \pm 0.38$ | $68.66 \pm 0.09$ | 6.8 |
| Ours | ✓ | $\mathbf{86.25 \pm 0.25}$ | $\mathbf{72.44 \pm 0.14}$ | $\mathbf{87.34 \pm 0.08}$ | $\mathbf{82.07 \pm 0.10}$ | $\underline{93.29 \pm 0.17}$ | $\underline{95.81 \pm 0.06}$ | $68.72 \pm 0.04$ | $\mathbf{1.4}$ |

The results show that our method maintains high accuracy across diverse graphs. Moreover, our method exhibits lower variance, demonstrating greater stability in its performance. When compared to other methods that employ learnable components, our method stands out. Notably, while AD-GCL is also a learnable method, it focuses only on learning the topology of the graph, which limits its effectiveness in many datasets. In contrast, our method leverages both the topology and attribute information, leading to more comprehensive and robust representation learning. As a result, our method consistently outperforms AD-GCL on most datasets, showcasing the advantages of augmenting both graph topology and node features.

On the large-scale ogbn-arxiv dataset, several methods encounter out-of-memory issues. While the supervised method GCN achieve the best results, our method performs almost equally well, with a very small margin of difference. This indicates that our approach is highly competitive when graph is large.

Table 2: Classification accuracy on graph classification datasets. The bold numbers represent the best results and the second highest results are underlined.

| Methods | NCI1 | MUTAG | PROTEINS | DD | RDT-B |
|---------|------|-------|----------|-----|-------|
| RU-GIN | $62.98 \pm 0.10$ | $87.61 \pm 0.39$ | $69.03 \pm 0.33$ | $74.22 \pm 0.30$ | $58.97 \pm 0.13$ |
| InfoGraph | $68.13 \pm 0.59$ | $87.71 \pm 1.77$ | $72.57 \pm 0.65$ | $\underline{75.23 \pm 0.39}$ | $78.79 \pm 2.14$ |
| GraphCL | $68.54 \pm 0.55$ | $88.29 \pm 1.31$ | $72.86 \pm 1.01$ | $74.70 \pm 0.70$ | $82.63 \pm 0.99$ |
| AD-GCL | $\mathbf{69.67 \pm 0.51}$ | $\underline{89.70 \pm 1.03}$ | $\underline{73.81 \pm 0.46}$ | $75.10 \pm 0.39$ | $\underline{85.52 \pm 0.79}$ |
| OURS | $\underline{69.35 \pm 0.63}$ | $\mathbf{89.71 \pm 0.73}$ | $\mathbf{73.21 \pm 0.40}$ | $\mathbf{75.51 \pm 0.57}$ | $\mathbf{83.99 \pm 0.34}$ |

### 5.3.2. GRAPH CLASSIFICATION RESULTS

In Table 2, we evaluate the performance of various graph classification methods across five datasets. We compare with four unsupervised/self-supervised learning baselines, which include randomly initialized untrained GIN (RU-GIN) (Xu et al., 2018), InfoGraph (Sun et al., 2020), GraphCL (You et al., 2020) and AD-GCL (Suresh et al., 2021a). They generally outperform in graph-level tasks. For all baselines, we report their performance based on AD-GCL (Suresh et al., 2021a). Our method achieves top or near-top performance on all datasets, demonstrating its robustness and ability to generalize across various tasks. Our method and AD-GCL (both learnable methods) perform better than the other methods across most datasets. However, Ours demonstrates superior generalization, particularly excelling on MUTAG and RDT-B, whereas AD-GCL performs slightly better on NCI1 and PROTEINS. Despite these slight differences, Ours achieves a balanced and consistent performance, providing strong evidence of its capability to handle diverse graph classification tasks effectively.

### 5.3.3. HETEROGENEOUS GRAPH RESULTS

In this section, we evaluate the performance of our method on node classification tasks using three heterogeneous graph datasets: Texas, Cornell, and Wisconsin (Pei et al., 2020). As summarized in Table 4, our approach consistently surpasses other methods on the Texas and Cornell datasets, achieving the best classification outcomes. This demonstrates the effectiveness of our model in capturing and leveraging the structural diversity inherent in heterogeneous graphs. On the Wisconsin dataset, our method delivers competitive results, ranking closely behind the top-performing

Table 3: Ablation study on the impact of feature and edge augmentation methods. In this table, the red values represent the highest classification accuracy for each dataset in each row, while the cells with a background indicate the highest performance in each column.

| Edge / Feature | Cora | | | PubMed | | | Wiki-CS | | |
|---|---|---|---|---|---|---|---|---|---|
| | Without Aug. | Random | Learnable | Without Aug. | Random | Learnable | Without Aug. | Random | Learnable |
| Without Aug. | $82.59 \pm 0.66$ | $84.40 \pm 0.99$ | $84.28 \pm 0.54$ | $85.11 \pm 0.19$ | $86.61 \pm 0.25$ | $86.81 \pm 0.23$ | $80.34 \pm 0.25$ | $81.05 \pm 0.23$ | $80.99 \pm 0.15$ |
| Random | $83.12 \pm 0.82$ | $85.84 \pm 0.44$ | $85.99 \pm 0.29$ | $84.97 \pm 0.14$ | $86.89 \pm 0.13$ | $87.07 \pm 0.25$ | $80.12 \pm 0.49$ | $81.61 \pm 0.19$ | $82.03 \pm 0.07$ |
| Learnable | $83.36 \pm 0.54$ | $85.83 \pm 0.37$ | $86.25 \pm 0.25$ | $85.37 \pm 0.07$ | $87.04 \pm 0.20$ | $87.34 \pm 0.08$ | $80.13 \pm 0.18$ | $81.51 \pm 0.14$ | $82.17 \pm 0.15$ |

Table 4: Classification accuracy on three heterogeneous datasets. The bold numbers represent the best results and the second highest results are underlined.

| Methods | Texas | Cornell | Wisconsin |
|---|---|---|---|
| GCN | $57.69 \pm 4.46$ | $48.44 \pm 4.82$ | $58.01 \pm 1.90$ |
| GAT | $56.19 \pm 2.16$ | $47.62 \pm 4.06$ | $55.12 \pm 3.23$ |
| DGI | $57.55 \pm 2.94$ | $47.89 \pm 2.05$ | $47.56 \pm 5.56$ |
| GMI | $48.98 \pm 3.60$ | $41.09 \pm 5.24$ | $51.24 \pm 3.93$ |
| GCA | $58.57 \pm 2.10$ | $47.45 \pm 0.88$ | $52.04 \pm 3.34$ |
| BGRL | $54.88 \pm 3.35$ | $35.15 \pm 10.43$ | $48.59 \pm 3.77$ |
| GREET | $62.79 \pm 2.57$ | $51.90 \pm 2.47$ | $63.11 \pm 0.74$ |
| SGRL | $60.00 \pm 0.27$ | $50.07 \pm 0.33$ | $56.92 \pm 0.24$ |
| GRACEIS | $61.90 \pm 2.40$ | $48.57 \pm 1.95$ | $59.10 \pm 0.96$ |
| AD-GCL | $60.82 \pm 2.49$ | $49.52 \pm 2.93$ | $57.01 \pm 2.70$ |
| JOAO | $62.45 \pm 2.96$ | $51.02 \pm 0.96$ | $60.90 \pm 0.87$ |
| Ours | $65.31 \pm 1.29$ | $53.06 \pm 1.05$ | $61.99 \pm 1.36$ |

approach. These findings highlight the strong generalization ability of our method across diverse heterogeneous graph structures.

## 5.4. Ablation Study ($Q2$)

### 5.4.1. EFFECT OF AUGMENTATION STRATEGIES

In this ablation study, We assess the influence of various augmentation strategies on classification accuracy across multiple datasets, comparing three approaches: no augmentation, random augmentation, and learnable augmentation. The results are detailed in the Table 3. Specifically, "without augmentation" refers to using the original data without any additional operations. "Random augmentation" involves randomly dropping nodes or edges according to a certain ratio, while "learnable augmentation" leverages our proposed learnable method, which allows for adaptive manipulation of node features and edge structures.

Overall, we observe that learnable augmentation consistently outperforms random augmentation, and random augmentation in turn performs better than the no augmentation approach. This pattern holds across most datasets, with both augmentation strategies improving performance in comparison to using the original data. However, the effects of augmenting edge and feature structures vary across datasets. For instance, on Cora and PubMed, augmenting the edge

structure yields a greater improvement than augmenting the node features. Compared to other datasets, WikiCS has a relatively higher number of average edges. The learnable augmentation method may not effectively enhance model performance when modifying the edge structure. Moreover, the node features in WikiCS have relatively low dimensionality. It indicates that edges may be more important than features in WikiCS. As a result, modifying features without adjusting edges might lead to suboptimal performance. Overall, in cases where random augmentation is applied, it can sometimes cause excessive disruption to the graph structure, especially on datasets where the edge relationships are more crucial to the performance. Therefore, our learnable augmentation method offers a more adaptive and data-aware alternative in preserving the underlying data distribution while improving performance.

Table 5: Performance comparison of GRACE and $Sp^2GCL$ before and after applying the proposed PiNGDA augmentation across benchmark datasets.

| Methods | Cora | CiteSeer | PubMed | Amazon-Photo | Coauthor-Phy |
|---|---|---|---|---|---|
| GRACE | $83.43 \pm 0.32$ | $70.93 \pm 0.21$ | $85.90 \pm 0.24$ | $93.13 \pm 0.17$ | $95.74 \pm 0.06$ |
| +PiNGDA | $84.28 \pm 0.24$ | $71.47 \pm 0.20$ | $86.79 \pm 0.27$ | $93.19 \pm 0.18$ | $95.81 \pm 0.05$ |
| $Sp^2GCL$ | $82.45 \pm 0.35$ | $65.54 \pm 0.51$ | $84.26 \pm 0.29$ | $93.05 \pm 0.23$ | $95.73 \pm 0.04$ |
| +PiNGDA | $83.89 \pm 0.48$ | $67.21 \pm 0.63$ | $84.73 \pm 0.23$ | $93.11 \pm 0.14$ | $95.74 \pm 0.04$ |

### 5.4.2. GENERALIZATION TO OTHER GCL MODELS

To evaluate the generalizability of our proposed noise augmentation method, we conducted an ablation study by extending it to other GCL frameworks. Specifically, we selected a classical GCL method (GRACE (Zhu et al., 2020)) and a more recent approach ($Sp^2GCL$ (Bo et al., 2023)) to validate the robustness and effectiveness of our approach when applied to diverse contrastive methods. As shown in Table 5, our method consistently improves performance across various datasets when integrated into both models. This highlights the generalizability and robustness of our method when incorporated into various contrastive learning frameworks.

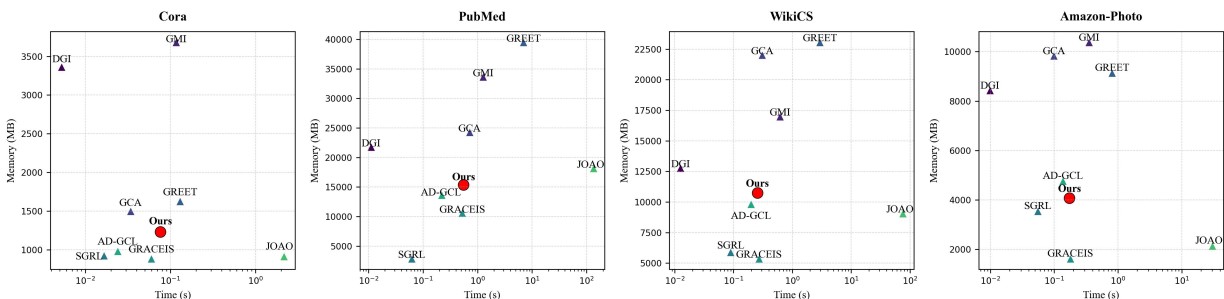

Figure 2: Comparison in terms of training time of one epoch, and memory costs between different graph representation learning methods. On Amazon-Photo, the model is trained with a batch size of 256 due to the memory limit.

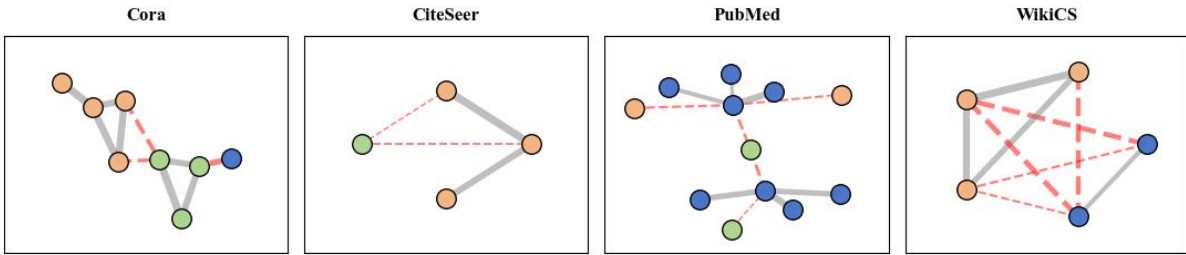

Figure 3: Visualization of noise in graph data. Nodes are selected from the graph which are colored according to their labels, edges are colored to differentiate between intra-class (solid gray) and inter-class (dashed red) relationships. The node layout is determined by their connectivity relationships. The width of the edges corresponds to the learned weights.

### 5.5. Efficiency Analysis ($Q3$)

Figure 2 reports the time and memory efficiency of our learnable method compared to other methods across multiple datasets. Compared with some GCL methods, our method has an advantage in occupying less memory. Meanwhile, in terms of runtime, our method is not significantly different from other methods. While some more complex methods may marginally outperform our approach in terms of computational cost, they come at a poor performance on accuracy. On the other hand, our learnable augmentation method offers an optimal trade-off between accuracy and computational efficiency.

### 5.6. Topological Noise Visualization ($Q4$)

Although our method introduces noise to both edges and node attributes, we only visualize the topological (edge-based) component here. This is because node attributes are typically high-dimensional and lack an inherent spatial structure, making them difficult to visualize intuitively. In Figure 3, we visualize the effects of topological noise on graph through edge weights. Specifically, nodes are selected from the full graph and different classes are represented using distinct colors. The node layout is determined by their connectivity relationships. Edges connecting nodes of the same class are shown as solid gray lines, while edges linking nodes of different classes are displayed as dashed red

lines. The width of the edges in the figure corresponds to the learned weights, which are indicative of the strength of the connections between nodes. It highlights the trend of our method towards removing inter-class edges while maintaining intra-class connections. By focusing on the removal of inter-class edges and preserving intra-class edges, our method ensures that the most relevant class-specific information is retained. It leads to more reliable augmentations.

## 6. Conclusion

In this paper, we propose Pi-Noise driven Graph Data Augmentation (PiNGDA), a novel method for graph data augmentation in graph contrastive learning (GCL). PiNGDA leverages the concept of positive-incentive noise to mitigate the instability commonly observed in traditional augmentation methods by learning adaptive noise. Furthermore, we introduce the notion of task entropy for classical GCL, showing that standard GCL can be viewed as an approximation under the optimization framework of positive-incentive noise. By employing a noise generator that learns beneficial noise, PiNGDA achieves performance improvements across various graph tasks. Additionally, future work may explore alternative strategies for incorporating positive-incentive noise with diverse noise distributions.

## Impact Statement

This paper presents work whose goal is to advance the field of Machine Learning. There are many potential societal consequences of our work, none which we feel must be specifically highlighted here.

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

# Appendix

## A. Derivation of Eq. (12)

We can expand the density of $\mathcal{N}(0, \kappa_{\boldsymbol{\theta}}(\boldsymbol{u}, \boldsymbol{\varepsilon})^{-1})$ and substitute it into $\mathcal{L}$,

$$\mathcal{L} = \frac{1}{n} \sum_{\boldsymbol{u}} \left( \log C + \frac{1}{2} \log \kappa_{\boldsymbol{\theta}}(\boldsymbol{u}, \boldsymbol{\varepsilon_0}) - \frac{1}{2} \right), \tag{23}$$

where the details can be found in Appendix A as

$$p(\alpha|\boldsymbol{u}, \boldsymbol{\varepsilon}_0) = C\sqrt{\kappa_{\boldsymbol{\theta}}(\boldsymbol{u}, \boldsymbol{\varepsilon}_0)} \exp(-\frac{\alpha^2}{2} \cdot \kappa_{\boldsymbol{\theta}}(\boldsymbol{u}, \boldsymbol{\varepsilon}_0))$$
$$\Longrightarrow \log p(\alpha|\boldsymbol{u}, \boldsymbol{\varepsilon}_0) = \log C + \frac{1}{2} \log \kappa_{\boldsymbol{\theta}}(\boldsymbol{u}, \boldsymbol{\varepsilon}_0) - \frac{\alpha^2}{2}\kappa_{\boldsymbol{\theta}}(\boldsymbol{u}, \boldsymbol{\varepsilon}_0). \tag{24}$$

Substituting it into $\mathcal{L}$, we can simplify $\mathcal{L}$ as

$$\begin{aligned}
\mathcal{L} &= \frac{1}{n} \sum_{\boldsymbol{u}} \int p(\alpha|\boldsymbol{u}, \boldsymbol{\varepsilon_0}) \cdot \left( \log C + \frac{1}{2} \log \kappa_{\boldsymbol{\theta}}(\boldsymbol{u}, \boldsymbol{\varepsilon_0}) - \frac{\alpha^2}{2} \cdot \kappa_{\boldsymbol{\theta}}(\boldsymbol{u}, \boldsymbol{\varepsilon_0}) \right) d\alpha \\
&= \frac{1}{n} \sum_{\boldsymbol{u}} \left( \log C + \frac{1}{2} \log \kappa_{\boldsymbol{\theta}}(\boldsymbol{u}, \boldsymbol{\varepsilon_0}) - \frac{\kappa_{\boldsymbol{\theta}}(\boldsymbol{u}, \boldsymbol{\varepsilon_0})}{2} \int \alpha^2 p(\alpha|\boldsymbol{u}, \boldsymbol{\varepsilon_0}) d\alpha \right) d\alpha \\
&= \frac{1}{n} \sum_{\boldsymbol{u}} \left( \log C + \frac{1}{2} \log \kappa_{\boldsymbol{\theta}}(\boldsymbol{u}, \boldsymbol{\varepsilon_0}) - \frac{1}{2} \right),
\end{aligned} \tag{25}$$

where $C$ in the above equation represents a constant independent of learnable parameters. In the final step, we use the fact that $\int \alpha^2 p(\alpha|\boldsymbol{u}, \boldsymbol{\varepsilon}_0) d\alpha = \kappa_{\boldsymbol{\theta}}(\boldsymbol{u}, \boldsymbol{\varepsilon}_0)^{-1}$.

## B. More Experimental Details

### B.1. Datasets

We use seven benchmark datasets for semi-supervised node classification, including Cora, Citeseer, Pubmed (Sen et al., 2008), Wiki-CS (Mernyei & Cangea, 2020), Amazon-Photo, Coauthor-CS (Shchur et al., 2019) and ogbn-arxiv (Hu et al., 2020). The detailed statistics of the datasets are summarized in Table 6. For all datasets, we randomly split the datasets, where 10%, 10%, and the rest 80% of nodes are selected for the training, validation, and test set, respectively.

We evaluate our proposed framework in the semi-supervised learning setting on graph classification on the benchmark TUDataset (Morris et al., 2020). The detailed statistics of the datasets are summarized in Table 7.

Table 6: Statistics of datasets used in node classification experiments.

| Dataset | Nodes | Edges | Features | Classes |
|---|---|---|---|---|
| Cora | 2,708 | 5,429 | 1,433 | 7 |
| Citeseer | 3,327 | 4,732 | 3,703 | 6 |
| PubMed | 19,717 | 44,338 | 500 | 3 |
| Wiki-CS | 11,701 | 216,123 | 300 | 10 |
| Amazon-Photo | 7,650 | 119,081 | 745 | 8 |
| Coauthor-CS | 18,333 | 81,894 | 6,805 | 15 |
| ogbn-arxiv | 169,343 | 1,166,243 | 128 | 40 |

### B.2. Implementation Details

Our experiments are conducted on an NVIDIA 4090 GPU (24 GB memory) for most datasets and on an NVIDIA A100 GPU (40 GB memory) for OGB-arxiv. For our proposed method, we employ a two-layer GCN network with PReLU activation, where the hidden layer dimension is set to 512, and the final embedding dimension is 256. Additionally, we utilize a projection head, consisting of a 256-dimensional fully connected layer with ReLU activation, followed by a 256-dimensional

Table 7: Statistics of datasets used in graph classification experiments.

| Dataset | Graphs | Avg. Nodes | Avg. Edges | Classes |
|---------|--------|------------|------------|---------|
| NCI1 | 4110 | 29.87 | 32.3 | 2 |
| MUTAG | 188 | 39.06 | 72.82 | 2 |
| PROTEINS | 1113 | 39.06 | 72.82 | 2 |
| DD | 1178 | 284.32 | 715.66 | 2 |
| RDT-B | 2000 | 429.6 | 497.75 | 2 |

linear layer. Edge noise generator uses an MLP to process node features and then applies Gumbel-Softmax sampling method. Feature noise generator uses two MLPs to estimate the mean and variance of feature noise and then uses the reparameterization trick.

### B.3. Ablation Study on Graph Classification

Table 8: Ablation study on the impact of feature and edge augmentation methods. In this table, the red values represent the highest classification accuracy for each dataset in each row, while the cells with a background indicate the highest performance in each column.

| Edge / Feature | PROTEINS | | | DD | | | NCI1 | | |
|---|---|---|---|---|---|---|---|---|---|
| | Without Aug. | Random | Learnable | Without Aug. | Random | Learnable | Without Aug. | Random | Learnable |
| Without Aug. | $72.49 \pm 0.90$ | $71.56 \pm 0.41$ | $73.16 \pm 0.77$ | $75.24 \pm 0.21$ | $73.07 \pm 0.52$ | $75.53 \pm 0.76$ | $68.59 \pm 0.75$ | $67.40 \pm 1.24$ | $68.83 \pm 0.69$ |
| Random | $72.38 \pm 0.46$ | $72.38 \pm 0.79$ | $72.49 \pm 0.58$ | $75.06 \pm 0.88$ | $74.23 \pm 0.82$ | $74.80 \pm 0.46$ | $69.05 \pm 0.74$ | $68.54 \pm 1.02$ | $68.84 \pm 0.65$ |
| Learnable | $72.45 \pm 1.24$ | $72.15 \pm 0.37$ | $\mathbf{73.21 \pm 0.40}$ | $75.16 \pm 0.71$ | $74.83 \pm 0.46$ | $\mathbf{75.70 \pm 0.42}$ | $69.11 \pm 0.62$ | $67.93 \pm 1.22$ | $\mathbf{69.35 \pm 0.63}$ |

We also explore the impact of different augmentation on graph classification performance across three datasets: PROTEINS, DD, and NCI1. Overall, the effectiveness of augmentation strategies varies across datasets, largely due to the differing semantic roles that node attributes play in each graph. In some cases, neither random nor learnable perturbations alone can fully capture the optimal augmentation pattern. Nevertheless, our results show that the combination of learnable feature and edge augmentations consistently yields strong and stable performance across all datasets. This demonstrates the robustness and generalizability of our augmentation framework, especially in adapting to diverse graph structures and attribute types.

### B.4. Hyperparameter Analysis

| Datasets | epoch | lr | wd | $\tau$ |
|----------|-------|-----|-----|-----|
| Cora | 500 | 5e-4 | 1e-4 | 0.3 |
| CiteSeer | 500 | 5e-4 | 1e-4 | 0.3 |
| PubMed | 1000 | 1e-3 | 1e-4 | 0.3 |
| WikiCS | 1500 | 5e-4 | 1e-4 | 0.3 |
| Amazon-Photo | 2000 | 1e-2 | 1e-4 | 0.3 |
| Coauthor-Phy | 2000 | 1e-2 | 1e-4 | 0.5 |
| ogbn-arxiv | 500 | 1e-3 | 1e-4 | 0.3 |

Table 9: Hyper-parameters that vary for different datasets.

Some hyper-parameters of the experiment vary on different datasets, which is shown in Table 9. For the learnable noise generators, we use separate optimizers with learning rates of 0.0001 for edges and 0.001 for features, and apply a weight decay of 0.0001 to both. Specifically, we carry out grid search for the hyper-parameters on the following search space:

- Number of training epochs: $\{500, 1000, 1500, 2000, 3000\}$.

- Learning rate for training: $\{1e-2, 1e-3, 5e-4\}$

- Weight decay for training: $\{1e-3, 1e-4\}$

- Random Edge dropping rates $p_{e1}, p_{e2}$: $\{0.1, 0.2, 0.3, 0.4, 0.5, 0.6\}$

- Random Feature masking rates $p_{f1}, p_{f2}$: $\{0.1, 0.3, 0.5\}$

- Temperature $\tau$: $\{0.3, 0.4, 0.5\}$

For each dataset, a set of hyper-parameters is chosen to obtain the best average accuracy. Figures 4 and 5 show the results

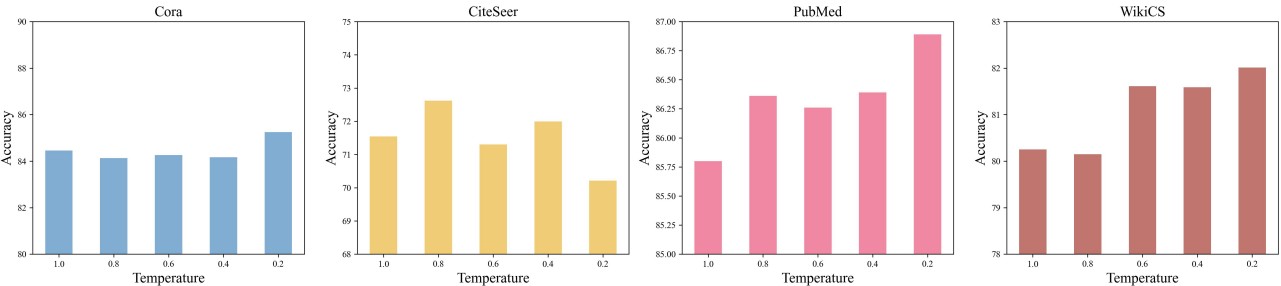

Figure 4: The effect of the temperature $\tau$.

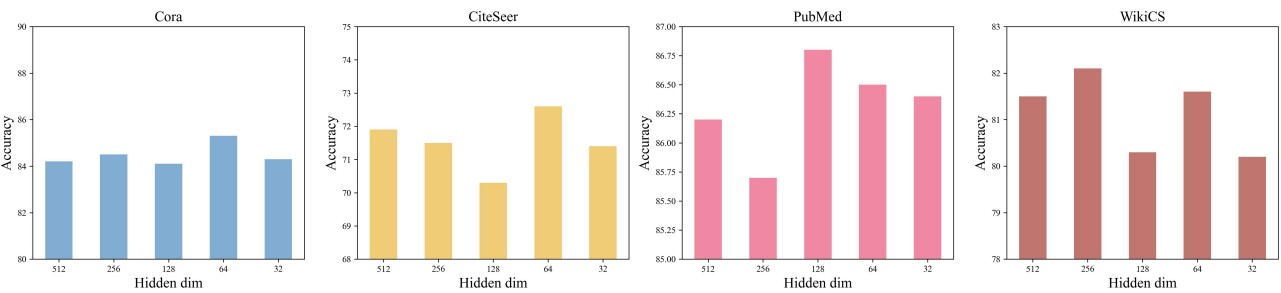

Figure 5: The effect of the hidden dim.

of node classification with variable temperature $\tau$ and hidden dim. We can observe that our PiNGDA is not very sensitive to temperature and hidden dim. For small graphs, smaller temperature leads to better performance.

