# OpenReview forum: "Learn Beneficial Noise as Graph Augmentation"
_ICML.cc/2025/Conference — ICML 2025 poster_

### Official Review · Reviewer_iJqw · 2025-03-07

**Overall Recommendation:** 4

**Summary:**

The paper proposes a graph contrastive learning method called Positive-incentive Noise driven Graph Data Augmentation (PiNGDA), which makes the model learn to generate perturbations that benefit the training. Comprehensive experiments are conducted to evaluate the performance of the method.

**Claims And Evidence:**

Yes.

The claims of the effectiveness and efficiency of the method are supported by experiments.
The theoretical claim that "the standard GCL with pre-defined augmentations is equivalent to estimate the beneficial noise via the point estimation" is proved with mathematical derivation based on information theory.

**Essential References Not Discussed:**

No.

**Experimental Designs Or Analyses:**

All the experimental designs and analyses are checked. Comprehensive experiments are conducted to evaluate the proposed method's performance.

However, the detailed hyperparameter of the training is not listed.

Moreover, as is mentioned in the caption of Figure 2, the model can be trained with a batch size to decrease memory burden, which is not discussed in Table. 1, where some methods report OOM on some datasets.

**Methods And Evaluation Criteria:**

Yes.

The performance of the proposed PiNGDA, as a GCL method, is evaluated in several aspects, including node classification, graph classification, node classification on heterogenous graphs, efficiency, visualization, and ablation study.

**Other Comments Or Suggestions:**

No.

**Other Strengths And Weaknesses:**

S1: According to the results of experiments, the effectiveness and stability of the method are validated comprehensively.

S2: According to the visualization, the proposed method tends to remove inter-class edges while maintaining intra-class edges, which is a common strategy applied in supervised learning.

W1&W2: As is mentioned in 'Experimental Designs or Analyses', the detailed hyperparameter of the training is not listed. Moreover, as is mentioned in Figure 2, the model can be trained with a batch size to decrease memory burden, which is not discussed in Table 1, where some methods report OOM on some datasets.

**Questions For Authors:**

Major questions: Please see weaknesses.

Here are some tiny questions:

1. JOAO is mentioned as 'learnable' in Table 1. However, to the best of my knowledge, JOAO randomly selects augmentation methods, which are all randomly applied. The projection head of each augmentation method differs.

2. The further discussion of 2.3 is placed in Appendix C after the detailed discussion of 3.3 and 4.2, which is kind of disordered.

3. The performance of the method on Wiki-CS in Table 2 is different from that in Table 1, why? The difference also appears between Table 7 and Table 3.

4. The results of the ablation study are kind of hard to read. Maybe the average value of columns and rows can be added to better show the trend of difference between w/o Aug., Random, and Learnable.

5. Can the attribute noise also be visualized?

**Relation To Broader Scientific Literature:**

The paper proposes a novel method that learns to generate augmented views, other than the adversarial method (AD-GCL). With a similar computational and time burden to AD-GCL, the proposed method achieves better performance than AD-GCL on most node classification and graph classification tasks, including higher accuracy and lower standards, indicating its effectiveness and stability.

**Theoretical Claims:**

Yes.

An auxiliary variable is introduced to prove the theoretical claim (Eq. (13)). The proof is correct and rigorous.

---

> ### Author Rebuttal · Authors · 2025-03-31
>
> **Response to Reviewer iJqw**
>
> We greatly thank you for the detailed and valuable comments. Please find our responses to the comments as follows:
>
> >**W1&W2:** As is mentioned in 'Experimental Designs or Analyses', the detailed hyperparameter of the training is not listed. Moreover, as is mentioned in Figure 2, the model can be trained with a batch size to decrease memory burden, which is not discussed in Table 1, where some methods report OOM on some datasets.
>
> **Reply:** Thank you for your valuable feedback. We will add detailed hyperparameter settings to improve clarity and reproducibility. Regarding batch size and memory efficiency, all **GCL methods were trained using the same batch size** to ensure a **fair comparison**. For smaller datasets, we do not apply batch processing, for large datasets, we set the batch size of 256. Our experiments are conducted on an NVIDIA 4090 GPU (24 GB memory) for most datasets and on an NVIDIA A100 GPU (40 GB memory) for OGB-arxiv. However, we recognize that this was not explicitly stated in Table 1. We will update Table 1 with a clarification that all methods follow the same batch size setting and discuss the impact of batch size adjustments on memory consumption. We appreciate your insightful comments and will revise the paper accordingly.
>
> >**Q1:** JOAO is mentioned as 'learnable' in Table 1. However, to the best of my knowledge, JOAO randomly selects augmentation methods, which are all randomly applied. The projection head of each augmentation method differs.
>
> **Reply:** Thank you for your insightful comment. While **JOAO does not learn augmentations directly**, it does **adaptively select** augmentation strategies based on a similarity-based policy. Referring to it as "learnable" in Table 1 may not be the most accurate description. Instead, "adaptable" better captures its mechanism. We will revise it in the paper and provide a clearer explanation of this distinction.
>
> >**Q2:** The further discussion of 2.3 is placed in Appendix C after the detailed discussion of 3.3 and 4.2, which is kind of disordered.
>
> **Reply:** We appreciate your feedback regarding the structure of the paper.  We will reorganize the content to ensure a more intuitive and coherent presentation. Thank you for your valuable suggestion!
>
> >**Q3:** The performance of the method on Wiki-CS in Table 2 is different from that in Table 1, why? The difference also appears between Table 7 and Table 3.
>
> **Reply:** Thank you for your question. The differences in performance  arise because Table 1 and Table 3 report the best results, while in the ablation study, we re-run the experiments to ensure a **fair comparison** across different settings. The results in the ablation study reflect newly obtained experimental outcomes under controlled conditions.
>
> >**Q4:** The results of the ablation study are kind of hard to read. Maybe the average value of columns and rows can be added to better show the trend of difference between w/o Aug., Random, and Learnable.
>
> **Reply:** Thank you for your helpful suggestions. We will update Table 2 to better highlight the best results for improved clarity.
>
> >**Q5:** Can the attribute noise also be visualized?
>
> **Reply:** Thank you for your insightful suggestion. Similar to our response to  Reviewer PRCX, visualizing **attribute noise** is challenging because **node representations are high-dimensional** and lack the structured spatial relationships found in visual data. Even if represented as a **matrix or heatmap**, the absence of spatial correlations between adjacent values could lead to **misleading interpretations**.

---

### Official Review · Reviewer_J7B2 · 2025-03-10

**Overall Recommendation:** 4

**Summary:**

This paper proposes a framework named Positive-incentive Noise driven Graph Data Augmentation (PiNGDA). It theoretically analyzes the drawbacks of the existing data augmentation in GCL and leverages a π-noise generator to learn beneficial noise as the augmentations for GCL. Meanwhile, they also design a differentiable algorithm to efficiently generate the noise. From the experimental results, PiNGDA achieves the highest performance compared to the current baselines.

**Claims And Evidence:**

Yes. The authors design theoretical analyses of the proposed model and extensive experiments validate performance.

**Essential References Not Discussed:**

No.

**Experimental Designs Or Analyses:**

Yes. I have checked the experimental details in Section 5 and Appendix D. The results can validate performance.

**Methods And Evaluation Criteria:**

Yes. The proposed PiNGDA can quantify to learn the beneficial graph augmentations not random drop some nodes.

**Other Comments Or Suggestions:**

No.

**Other Strengths And Weaknesses:**

## Strengths:
1) The authors propose an interesting framework to directly learn the beneficial noise augmentations. After all, the noise generally is regarded as the harmful signal.
2) The theoretical analysis is logical, and the paper is well-organized to follow.

## Weaknesses:
1) The relationships between the proposed model and the learnable methods are not discussed in Section 1.
2) The graph contrastive learning task should be introduced in more detail in Section 1.
3) In contributions, the authors should clarify the merits of the proposed model from the Pi-Noise perspective.
4) The connections and differences between PINGDA and some current GCL models based on learnable strategies should be discussed in details.
5) The authors could add the ablation study such as extending the proposed noise augmentations to more current GCL models.
6) Apart from the Memory-Times experiments, Memory-Performance or FLOPs-Performance should be introduced to validate efficiency.
7) There are some typos. For example, in line 19, “. Where” should be “, where”. In Figure 3, it lacks explanations about the dashed line between two nodes.

**Questions For Authors:**

I hope the authors can state the generalization of the proposed model. For example, does it improve the performance on other new GCL models or graph tasks? (Please see Weakness 6)

**Relation To Broader Scientific Literature:**

In this paper, the authors theoretically analyze the drawbacks of the existing data augmentations and propose a novel PiNGDA to differentiable learn the beneficial graph augmentations.

**Theoretical Claims:**

Yes. I have checked the theoretical proofs in Section 3 and Appendix A. They are correct.

---

> ### Author Rebuttal · Authors · 2025-03-31
>
> **Response to Reviewer J7B2**
>
> We greatly thank you for the detailed and valuable comments. Please find our responses to the comments as follows:
>
> >**W1&2&3&4:** The relationships between the proposed model and the learnable methods are not discussed in Section 1. The graph contrastive learning task should be introduced in more detail in Section 1. The connections and differences between PINGDA and some current GCL models based on learnable strategies should be discussed in details. In contributions, the authors should clarify the merits of the proposed model from the Pi-Noise perspective.
>
> **Reply:** Thank you for your valuable suggestions. Due to space limitations, we did not provide an in-depth discussion of these aspects in Section 1.  We will provide a more comprehensive introduction to the graph contrastive learning task in the revised version. Additionally, we will provide a more detailed introduction to the graph contrastive learning task to improve clarity and comprehensiveness. For the specific differences  with AD-GCL, please refer to the response to **Reviewer PRCX**.
>
> >**W5:** The authors could add the ablation study such as extending the proposed noise augmentations to more current GCL models.
>
> **Reply:** We have included the requested ablation study by extending our proposed noise augmentations to more GCL models. Specifically, we selected both a classical GCL model (GRACE [1]) and a more recent approach (Sp²GCL [2]) to demonstrate the effectiveness of our method. The results are summarized in the table below:
>
> |               | Cora         | CiteSeer     | PubMed       | Amazon-Photo | Coauthor-Phy |
> | ------------- | ------------ | ------------ | ------------ | ------------ | ------------ |
> | GRACE         | 83.43 ± 0.32 | 70.93 ± 0.21 | 85.90 ± 0.24 | 93.13 ± 0.17 | 95.74 ± 0.06 |
> | GRACE+PiNGDA  | 84.28 ± 0.24 | 71.47 ± 0.20 | 86.79 ± 0.27 | 93.19 ± 0.18 | 95.81 ± 0.05 |
> | Sp²GCL        | 82.45 ± 0.35 | 65.54 ± 0.51 | 84.26 ± 0.29 | 93.05 ± 0.23 | 95.73 ± 0.04 |
> | Sp²GCL+PiNGDA | 83.89 ± 0.48 | 67.21 ± 0.63 | 84.73 ± 0.23 | 93.11 ± 0.14 | 95.74 ± 0.04 |
>
> These results highlight the impact of our augmentation approach across different GCL models.
>
> [1]Zhu, Yanqiao, Yichen Xu, Feng Yu, Qiang Liu, Shu Wu, and Liang Wang. "Deep graph contrastive representation learning." *arXiv preprint arXiv:2006.04131* (2020).
>
> [2]Bo, Deyu, Yuan Fang, Yang Liu, and Chuan Shi. "Graph contrastive learning with stable and scalable spectral encoding." *Advances in Neural Information Processing Systems* 36 (2023): 45516-45532.
>
> >**W6:** Apart from the Memory-Times experiments, Memory-Performance or FLOPs-Performance should be introduced to validate efficiency.
>
> **Reply:** The detailed tables are shown below. The results show how our method compares to existing baselines in terms of both computational cost and accuracy improvements. A negative value in ΔFLOPs or ΔMem. indicates **a reduction of our method** in computational cost or memory usage, while a positive value in ΔAcc% reflects our accuracy improvement over the respective method.
>
> | Methods | Cora ΔFLOPs (G) | ΔAcc% | PubMedΔFLOPs (G) | ΔAcc% | WikiCSΔFLOPs (G) | ΔAcc% | Amazon-Photo ΔFLOPs (G) | ΔAcc% |
> |-|-|-|-|-|-|-|-|-|
> | DGI| -3.27| +3.60 | -4.93| +2.61 | -0.50| +3.44 | -3.83| +4.81 |
> | GMI| -1.99| +3.17 | -5.06| +3.83 | -1.77| +2.60 | -2.93| +3.51 |
> | GCA | -0.71| +1.23 | -5.18| +0.31 | -3.04| +1.00 | -2.01| +0.11 |
> | BGRL| +0.33| +2.92 | +2.46| +2.03 | +1.45| +0.85 | +0.95| +0.11 |
> | GREET| -4.79| +7.18| -16.59| +1.48 | +0.72| +2.69 | -16.60| -0.34 |
> | GRACEIS | +0.53| +1.99 | +3.91| +3.30 | +2.33| +3.45 | +1.50| +2.22 |
>
> | Methods | Cora ΔMem. | ΔAcc(%) | PubMed ΔMem. | ΔAcc(%) | WikiCS ΔMem. | ΔAcc(%) | Amazon-Photo ΔMem. | ΔAcc(%) |
> |-|-|-|-|-|-|-|-|-|
> | DGI| -63.30%| +3.61%| -29.50%| +2.61%| -15.90%| +3.44%| -51.70%| +4.81%  |
> | GMI| -66.50%| +3.17%| -54.40%| +3.83%  | -36.70%| +2.60%  | -60.80%| +3.50%  |
> | GCA| -17.60%| +1.23%| -36.90%| +0.31%  | -51.20%| +0.99%| -58.60%| +0.11%|
> | GREET| -24.20%| +7.19%  | -61.10%| +1.48%  | -53.40%| +2.70%  | -55.50%| -0.34%  |
> | AD-GCL| +25.1%| +2.83%  | +12.6%| +3.69%  | +9.3%| +1.86%  | -14.60%| +1.49%  |
>
> Our results demonstrate that while some methods, such as GRACEIS and AD-GCL, achieve efficiency gains by significantly reducing FLOPs and memory usage, our approach **strikes a balance between efficiency and performance**.
>
> >**W7:** There are some typos. For example, in line 19, “. Where” should be “, where”. In Figure 3, it lacks explanations about the dashed line between two nodes.
>
> **Reply:** Thank you for your careful review. We will correct the typos. Regarding the dashed lines in Figure 3, they represent connections between nodes of different classes. We apologize for the unclear representation and will revise the figure or improve the explanation to ensure better clarity. We appreciate your valuable feedback and will make the necessary improvements.

---

### Official Review · Reviewer_smN2 · 2025-03-10

**Overall Recommendation:** 4

**Summary:**

This paper proposes a graph data augmentation method based on beneficial noise. The noise generator learns the optimal perturbation of graph structure and node features to solve the problem of insufficient stability of traditional data augmentation strategies in graph contrastive learning. Experimental verification shows that PiNGDA outperforms baseline methods in tasks.

**Claims And Evidence:**

The paper presents experimental results to support its main claims. The authors verified the method's effectiveness on node classification and graph classification datasets. Additionally, an ablation study on the key noise generation module was conducted, enhancing understanding of the key module's contribution. Thus, the proposed conclusions are somewhat powerful and credible.

**Essential References Not Discussed:**

No.

**Experimental Designs Or Analyses:**

The experimental settings are reasonable. The inclusion of ablation studies further strengthens the analysis by isolating the contributions of different components.

**Methods And Evaluation Criteria:**

This paper presents a new augmentation method for graph contrastive learning. The proposed approach effectively tackles the recognized issues by integrating learnable augmentations into the contrastive learning framework. In contrast to traditional augmentation techniques like node and edge dropping, this method enriches representation learning through the introduction of beneficial noise.

The chosen evaluation criteria, which consist of standard benchmarks for node classification and graph classification, are appropriate for assessing the effectiveness and generalization ability of the proposed method. The employment of multiple evaluation metrics guarantees a comprehensive analysis of the model's performance.

**Other Comments Or Suggestions:**

Please refer to the strengths and weaknesses.

**Other Strengths And Weaknesses:**

Strengths:
1. The main conclusion of the theoretical part is convincing.
2. The experiments are somehow abundant, covering both node classification and graph classification tasks.
3. The inclusion of ablation studies provides evidence for the effectiveness of different components of the method.

Weaknesses:

1. The theoretical analysis in Section 3.3 is interesting, but the definition of task entropy are unclear.
2. The paper does not detail the hyperparameters used in the experiment, such as learning rate batch size, etc., which will affect the reproducibility of the research.

**Questions For Authors:**

1. The theoretical analysis in Section 3.3 is intriguing. However, I am unclear about the definition and role of task entropy. Could you provide a more detailed explanation or an example to clarify its significance?
2. While the experimental setup is well-documented, the hyperparameter settings are not explicitly detailed. Could you provide more information on these aspects, either in the main text or in supplementary materials?
3. The computational efficiency in Figure 2 does not show much advantage of PiNGDA. Can you explain why this is the case and provide analyze of the complexity?

**Relation To Broader Scientific Literature:**

Previous graph contrastive augmentation methods included edge dropping and node dropping. Later GCL methods were developed based on these by introducing learnable dropping techniques.  This paper further expands on these approaches. It incorporates beneficial noise into contrastive learning, integrates these techniques into a unified framework, and proposes a new method.

**Theoretical Claims:**

The main theoretical assertions proposed in the paper have been proved mathematically. The author lists the assumptions and derives the conclusion. The derivation process is presented clearly.

---

> ### Author Rebuttal · Authors · 2025-03-31
>
> **Response to Reviewer smN2**
>
> We greatly thank you for the detailed and valuable comments. Please find our responses to the comments as follows:
>
> >**W1&Q1:** The theoretical analysis in Section 3.3 is interesting, but the definition of task entropy are unclear.The theoretical analysis in Section 3.3 is intriguing. However, I am unclear about the definition and role of task entropy. Could you provide a more detailed explanation or an example to clarify its significance?
>
> **Reply:** In our framework, **task entropy is used to measure the difficulty of a task**. The key idea is that adding noise can help simplify the task, which is quantitatively measured by the task complexity, making it easier for the model to learn meaningful representations. This aligns with our theoretical analysis, where learned beneficial noise can lead to better performance. We hope this explanation can answer your question and we would like to further clarify it if the concern is still well-addressed.
>
> >**W2&Q2:** The paper does not detail the hyperparameters used in the experiment, such as learning rate batch size, etc., which will affect the reproducibility of the research. While the experimental setup is well-documented, the hyperparameter settings are not explicitly detailed. Could you provide more information on these aspects, either in the main text or in supplementary materials?
>
> **Reply:** Thank you for your helpful suggestions. The hyperparameters were selected within the following ranges: For smaller datasets, we do not apply batch processing, for large datasets, we set the batch size of 256. The number of epochs varies from 500 to 2000 depending on the dataset. The learning rate is set between 0.0005 and 0.01, while the weight decay is kept constant at 0.0001 across all datasets. The feature dropout rates range from 0.0 to 0.3, and the edge dropout rates  vary between 0.1 and 0.4. The temperature is typically set to 0.3 for most datasets, with an exception for Coauthor-Phy, where it is set to 0.5. This is a simple description, the detailed version will be added later.
>
> >**Q3:** The computational efficiency in Figure 2 does not show much advantage of PiNGDA. Can you explain why this is the case and provide analyze of the complexity?
>
> **Reply:** The computational efficiency of PiNGDA may not appear significantly advantageous in Figure 2 because our method involves adding noise to all data points, which introduces additional computations. In the two noise generation modules, since each edge and each node feature is calculated by MLP, the computational complexity of the noise module is $\mathcal{O}(\mathcal{E} \cdot d) + \mathcal{O}(N \cdot d)$, where N is the number of nodes, $\mathcal{E}$ is the number of edges in the graph, and d is the dimension of the node feature. However, this process is crucial for improving the robustness and generalization of the model. To provide a more comprehensive comparison, we have included detailed memory-performance  analyses in the tables. Please refer to response to **Reviewer J7B2**.

---

### Official Review · Reviewer_LdjD · 2025-03-10

**Overall Recommendation:** 3

**Summary:**

This paper proposes a novel method called PiNGDA for addressing the instability of traditional heuristic augmentation techniques in graph contrastive learning (GCL). The authors introduce the concept of π-noise, which is beneficial noise that reduces task complexity, and design a trainable noise generator to produce optimal perturbations in both graph topology and node attributes. Experimental results on graph benchmarks demonstrate superior performance compared to existing methods.

**Claims And Evidence:**

The authors claim that learning beneficial noise through a π-noise framework can provide a more reliable and stable augmentation strategy. This claim is supported by theoretical analysis and extensive experiments across multiple datasets.

**Essential References Not Discussed:**

1) Spectral Feature Augmentation for Graph Contrastive Learning and Beyond, in AAAI 23

2) Unified Graph Augmentations for Generalized Contrastive Learning on Graphs, in NeurIPS 24

3) Graph Adversarial Self-Supervised Learning, in NeurIPS 21

**Experimental Designs Or Analyses:**

Yes, I have checked the soundness of the experimental designs (including the compared methods and experimental setups) and analyses.

**Methods And Evaluation Criteria:**

The proposed PiNGDA consists of a π-noise generator and a contrastive learning module. The noise generator produces topological and attribute noise using a Gaussian auxiliary variable and reparameterization tricks to ensure differentiability. The model is evaluated using node classification accuracy and graph classification accuracy across several datasets.

**Other Comments Or Suggestions:**

1) The organization of the paper could be improved. For example, separating the discussion of existing research in Section 3 would enhance readability.

2) Minor grammatical issues, such as the lowercase "where" in line 19, should be corrected.

  3)  The authors should provide more details on the experiments to ensure reproducibility.

**Other Strengths And Weaknesses:**

**Strengths**

The ablation studies provide valuable insights into the impact of different augmentation strategies.


**Weaknesses**

1) The π-noise framework itself is not entirely novel, as it is based on existing research. The innovation lies in its application to GCL.

 2)  The paper lacks detailed descriptions of some experimental settings, such as hardware environment and encoder choices, which may hinder reproducibility.

3) Compared to AD-GCL, PiNGDA does not show advantages in terms of time and space efficiency.

**Questions For Authors:**

How does PiNGDA compare to other baselines that also focus on adaptive or learnable graph augmentations, such as those mentioned in the "Essential References Not Discussed" section?

**Relation To Broader Scientific Literature:**

1) The authors provide a novel perspective on GCL by analyzing it through the lens of the π-Noise framework. They elucidate why traditional random noise augmentation methods lead to unstable performance, which offers a fresh direction for future research in GCL.

2) The proposed model demonstrates extensive applicability, achieving superior performance in both node classification and graph classification tasks.

**Theoretical Claims:**

The authors theoretically analyze the relationship between π-noise and the training loss in GCL. They show that predefined augmentations in existing GCL models can be considered as point estimations of π-noise, which may not always be reliable. By learning π-noise directly, PiNGDA provides a more robust approach to graph augmentation.

---

> ### Author Rebuttal · Authors · 2025-03-31
>
> **Response to Reviewer LdjD**
>
> We greatly thank you for the detailed and valuable comments. Please find our responses to the comments as follows:
>
> >**W1:** Innovation of application of $\pi$-noise to GCL.
>
> **Reply:** Although $\pi$-noise has been explored in other fields, its adaptation for graph data and integration within the GCL framework offers a unique perspective. The core challenge of π-noise—**how to define task entropy—remains highly difficult**, especially for **graph data**. This is not merely incremental work, but rather a fundamental challenge, which forms the core contribution of our research. Our work achieves **unified augmentations on both graph topology and node features within a theoretical framework**, which prior GCL methods have not been able to accomplish. This novel application allows us to leverage noise in a way that improves the quality of graph representations.
>
> >**W2:** Lacks detailed experimental settings.
>
> **Reply:**  Thank you for your suggestion. Our experiments are conducted on an NVIDIA 4090 GPU (24 GB memory) for most datasets and on an NVIDIA A100 GPU (40 GB memory) for OGB-arxiv. For our proposed method, we employ a two-layer GCN network with PReLU activation, where the hidden layer dimension is set to 512, and the final embedding dimension is 256. Additionally, we utilize a projection head, consisting of a 256-dimensional fully connected layer with ReLU activation, followed by a 256-dimensional linear layer. Edge noise generator uses an MLP to process node features and then applies Gumbel-Softmax sampling method. Feature noise generator uses two MLPs to estimate the mean and variance of feature noise and then uses the reparameterization trick. We appreciate your feedback and will ensure these details are clearly presented in the revised version.
>
> >**W3:** Compared to AD-GCL, PiNGDA does not show advantages in terms of time and space efficiency.
>
> **Reply:** Our method indeed introduces some additional computational overhead because we have two augmentation modules—one for graph structure and one for node features. This results in a slight increase in memory and time consumption. However, as shown in the table, this increase is not substantial, and more importantly, our method achieves significantly better performance, demonstrating its effectiveness.
> |Methods | Cora Mem.(M)|Time(s)|Acc(%) | PubMed Mem.(M)|Time(s)|Acc(%)|WikiCS Mem.(M)|Time(s)|Acc(%)|Amazon-Photo Mem.(M)|Time(s)|Acc(%)|
> |-|-|-|-|-|-|-|-|-|-|-|-|-|
> |AD-GCL(aug on graph only) |986|0.03|83.88|13632|0.22|84.23|9836|0.20|80.57|4768|0.14|91.92|
> |Ours(aug on graph & fea)|1070|0.04|86.25|15108|0.44|87.34|9704|0.21|82.07|4072|0.13|93.29|
> |Δ|84|0.01|+2.8%|1476|0.22|+3.7%|-132|0.01|+1.9%| -696|-0.01|+1.5%|
>
>
> >**C1&2&3:**
>
> **Reply:** Thank you for your helpful suggestions. We will restructure Section 3 to improve readability by separating the discussion of existing research. Additionally, we will correct minor grammatical issues. To enhance reproducibility, we will also provide more details on the experimental settings. We appreciate your valuable feedback and will make the necessary revisions.
>
> >**Q:** How does PiNGDA compare to other baselines that also focus on adaptive or learnable graph augmentations, such as those mentioned in the "Essential References Not Discussed" section?
>
> **Reply:**
>
> | Method| CORA|CiteSeer| PubMed | Amazon-Photo| Coauthor-Phy|
> |-|-|-|-|-|-|
> | **GOUDA** | 82.25 ± 0.54| 70.25 ± 1.24| 85.82 ± 0.71| 89.61 ± 1.17| 94.54 ± 0.59 |
> | **GASSL** | 81.82 ± 0.79| 69.51 ± 0.84| 84.91 ± 0.57| 92.14 ± 0.23| 94.93 ± 0.21|
> | **Ours**  | **86.25 ± 0.25** | **72.44 ± 0.14** | **87.34 ± 0.08** | **93.29 ± 0.17** | **95.81 ± 0.06** |
>
> Since the official code for Paper 1 was not available, we replicated the results from the supplementary materials submitted with Papers 2 and 3 from OpenReview. The performance difference between the reported results and our replication could be due to the absence of specific hyperparameters in the papers. We base our results on our own replication.
>
> Also, as suggested by **Review J7B2**, we also added PiNGDA to different GCL backbones and achieved good results.
> || Cora| CiteSeer|PubMed| Amazon-Photo | Coauthor-Phy |
> |-|-|-|-|-|-|
> | **GRACE**| 83.43 ± 0.32 | 70.93 ± 0.21 | 85.90 ± 0.24 | 93.13 ± 0.17 | 95.74 ± 0.06 |
> | **GRACE+PiNGDA**  | **84.28 ± 0.24**| **71.47 ± 0.20**| **86.79 ± 0.27** | **93.19 ± 0.18**| **95.81 ± 0.05**|
> | **Sp²GCL**| 82.45 ± 0.35 | 65.54 ± 0.51 | 84.26 ± 0.29 | 93.05 ± 0.23 | 95.73 ± 0.04 |
> | **Sp²GCL+PiNGDA** | **83.89 ± 0.48**|**67.21 ± 0.63**| **84.73 ± 0.23** | **93.11 ± 0.14**| **95.74 ± 0.04**|
>
> Our method consistently outperforms existing approaches across all datasets, achieving significant improvements in accuracy. This demonstrates the effectiveness of our adaptive graph augmentation strategy, which better captures the underlying structure of the data and enhances model generalization.

---

> > ### Comment · Reviewer_LdjD · 2025-04-04
> >
> > Thank you for your response. I am satisfied that most of my concerns have been addressed, and I hope to see these contents included in the final paper. If so, I will increase my rating.

---

> > > ### Author Response · Authors · 2025-04-04
> > >
> > > Thank you for your valuable suggestions! We sincerely appreciate your guidance and will incorporate all feedback into the revised manuscript.

---

### Official Review · Reviewer_65ie · 2025-03-12

**Overall Recommendation:** 4

**Summary:**

This paper proposes a graph contrastive learning (GCL) methods, namely PINGDA, with a novel learnable graph augmentation. The learnable augmentation follows a new information theory framework, namely positive-incentive noise. The authors propose to view all augmentations as “noise” and thus design a new algorithm to add beneficial noise to both graph topology and node attributes. The experiments are run on several graph tasks, including node classification and graph classification.

**Claims And Evidence:**

Yes, the theoretical claims are well verified by the experiments. In most cases, the proposed PINGDA achieves the best results. In some cases, the method also achieves sub-optimal results. I notice that the stds are usually smaller than other GCL methods, which seems to prove that the proposed idea of using noise makes sense.

**Essential References Not Discussed:**

none

**Experimental Designs Or Analyses:**

I checked the experimental design and analyses. The results can well support the claims and conclusions.

**Methods And Evaluation Criteria:**

The proposed methods and evaluation criteria make sense for the problem. The experiments contain both node classification and graph classification. The datasets also contain both heterogeneous and homogeneous graphs.

**Other Comments Or Suggestions:**

See above.

**Other Strengths And Weaknesses:**

Strengths:

1. The paper proposes an interesting and new perspective for graph augmentation in GCL. The idea of viewing graph augmentation as noise seems new as far as I know. It can therefore unify the augmentation on graph topology and node attributes, which is an advantage compared with the existing GCL models with learnable augmentations.
2. PINGDA also focuses on how to augment node attributes while most existing GCL models focus on how to modify the graph and only utilize the simple perturbations on attribute augmentations.
3. The experimental results seem promising. The learnable augmentations seem to well support the theoretical analysis. The experiments contain node classification and graph classification. The datasets consist of both homogeneous graphs and heterogeneous graphs. I noticed that the standard derivations are smaller in most cases, which validates the effectiveness of the learnable augmentation.


Weaknesses:

1. The introduction of positive-incentive noise and the discussions with the existing papers of noise are not enough, especially in the main paper.
2. Figure 1 can be further improved. It is hard to distinguish how to simultaneously generate augmentations on both graph topology and node features. Meanwhile, $\mathcal{N}(0, \mathcal{I})$ in the figure is inconsistent with the notations appearing in the main paper.
3. In Table 2, the best results should be further highlighted. The current version is not intuitive.
4. The caption of Table 4 is quite confusing. The authors fail to clarify that the datasets are heterogeneous graphs in the caption.

**Questions For Authors:**

1. Why do the authors assume that the topological noise is a Bernoulli distribution?
2. I cannot completely understand the meaning of Figure 3. How are the nodes visualized? t-SNE? Why does it only contain dozens of nodes? As we all know, there are thousands of nodes in a graph. The authors should clarify this in the rebuttal and the main paper.
3. What’s the additional computational and space complexity? Will it lead to a significant burden on both computation and memory?

**Relation To Broader Scientific Literature:**

The paper provides a new perspective for stable graph augmentation, the theory is well grounded by information bottleneck.  It is also an interesting advantage compared with the existing GCL methods, such as GCA and JOAO.

**Theoretical Claims:**

I checked most of the mathematical derivations. It seems that the formulations are roughly correct.

---

> ### Author Rebuttal · Authors · 2025-03-31
>
> **Response to Reviewer 65ie**
>
> We greatly thank you for the detailed and valuable comments. Please find our responses to the comments as follows:
>
> >**W1:** The introduction of positive-incentive noise and the discussions with the existing papers of noise are not enough, especially in the main paper.
>
> **Reply:** Thank you for your valuable suggestion. Due to space limitations, we only provide a simple and brief discussion in the main paper.  We will consider expanding this discussion in the supplementary material or refining the main text to provide a clearer comparison.
>
> >**W2:** Figure 1 can be further improved. It is hard to distinguish how to simultaneously generate augmentations on both graph topology and node features. Meanwhile, $\mathcal{N}(0,I)$ in the figure is inconsistent with the notations appearing in the main paper.
>
> **Reply:** Thank you for your valuable feedback.  We will refine the figure to enhance clarity. As for the notation inconsistency, we apologize for any confusion. In **Appendix [B]**, we explicitly describe the reparameterization trick used in our method, where the covariance matrix is defined as $\Sigma = \text{diag}(\sigma^2)$, ensuring a diagonal structure. Here, $\sigma\sim \mathcal{N}(0, I)$, aligning with the notations in the figure.
>
> >**W3&4:** In Table 2, the best results should be further highlighted. The current version is not intuitive. The caption of Table 4 is quite confusing. The authors fail to clarify that the datasets are heterogeneous graphs in the caption.
>
> **Reply:** Thank you for your helpful suggestions. We will update Table 2 to better highlight the best results for improved clarity and we will revise the caption of Table 4 to clearly specify that the datasets are heterogeneous graphs.
>
> >**Q1:** Why do the authors assume that the topological noise is a Bernoulli distribution?
>
> **Reply:** Thank you for your question. We assume that the **topological noise follows a Bernoulli distribution** because the simplest way to model edge perturbation is to learn whether an edge should be kept or removed, which naturally aligns with a **0/1 binary decision process**. This makes Bernoulli distribution a **straightforward and effective choice** for modeling edge retention or deletion in graph structures.
>
> >**Q2:** I cannot completely understand the meaning of Figure 3. How are the nodes visualized? t-SNE? Why does it only contain dozens of nodes? As we all know, there are thousands of nodes in a graph. The authors should clarify this in the rebuttal and the main paper.
>
> **Reply:** Sorry for the lack of clarity in Figure 3. To improve visualization, the nodes shown in the figure are **a subset selected from thousands of nodes** for a case study. This selection helps make the visualization more interpretable. The node distribution in the figure is based on their **connectivity relationships**, rather than a dimensionality reduction method like t-SNE. We will clarify this in the main paper. Thank you for your valuable feedback.
>
> >**Q3:** What’s the additional computational and space complexity? Will it lead to a significant burden on both computation and memory?
>
> **Reply:** Thank you for your insightful question. The **computational and space complexity** of noise learning depends on the design of the noise network. In the two noise generation modules, since each edge and each node feature is calculated by MLP, the computational complexity of the noise module is $\mathcal{O}(\mathcal{E} \cdot d) + \mathcal{O}(N \cdot d)$, where N is the number of nodes, $\mathcal{E}$ is the number of edges in the graph, and d is the dimension of the node feature. To minimize the burden, we have carefully **simplified the network structure**, such as reducing the number of hidden layers. The memory overhead as shown in Figure 2, is not significant. While our method introduces some additional computational costs, it provides clear performance improvements, making the trade-off worthwhile. The detailed table can be found in the response to **Reviewer J7B2**. We appreciate your feedback and will clarify this further in the paper.

---

### Official Review · Reviewer_PRCX · 2025-03-12

**Overall Recommendation:** 3

**Summary:**

This work introduces the PiNGDA method, designed to enhance graph data augmentation through the incorporation of beneficial noise. The paper also introduces the concept of task entropy, offering a fresh lens through which to comprehend the objective function of contrastive learning. Practical results demonstrate that PiNGDA leads to performance gains, thereby confirming its efficacy.

**Claims And Evidence:**

This work asserts that its method addresses the stability issues associated with traditional methods. According to experimental findings, the proposed method excels in various tasks, including node classification and graph classification, demonstrating its efficacy. Additionally, ablation studies provide insight into the individual contributions of each module.

**Essential References Not Discussed:**

No

**Experimental Designs Or Analyses:**

The experimental design of this paper is well-conceived, encompassing various benchmark datasets to fortify the reliability of the experimental outcomes. It employs widely-acknowledged baseline methods for comparative analysis. Furthermore, ablation studies are included to dissect the individual impacts of different modules.

**Methods And Evaluation Criteria:**

This paper introduces a novel graph contrastive learning data augmentation technique along with an innovative framework aimed at resolving issues with existing approaches. By employing a learnable augmentation strategy, the framework enhances the model's capabilities while maintaining the integrity of graph structure information. To ensure fairness and comparability in experimental results, the evaluation criteria adhere to traditional GCL methods, utilizing standard contrastive learning loss functions and widely accepted evaluation metrics.

**Other Comments Or Suggestions:**

See the weaknesses.

**Other Strengths And Weaknesses:**

Strengths:
1）The paper is clearly expressed, allowing readers to easily understand the research content. In addition, the experimental section provides sufficient comparative analysis. The paper conducts experiments on multiple datasets and compares the performance of different methods, such an experimental design can help verify the robustness and applicability of the method.
2） The paper also provides a detailed analysis in the ablation experiment section to further verify the effectiveness of the method. It analyzes the impact of different components in the method on the final performance through detailed ablation experiments. This can help understand which parts contribute the most to the final result.
3） The authors visualize the changes in edge weights to make the abstract “noise” expression more intuitive. This helps us understand how the model adjusts the graph structure, optimizes information propagation, improving the interpretability of the method.

Weaknesses:
1) The depth of the discussion of the experimental results can be further improved, such as the explanation of certain experimental phenomena and the possible differences between different datasets. For example, in the Table 2, the learnable methods seem not to perform as well as the other two datasets.
2) The mathematical derivation in the method section can be more detailed to enhance readability and understanding. For example, the random variable alpha is a bit confusing. Why is this variable necessary? And why do you assume the probability distribution showed in equation 4? It will be very helpful if they are explained clearly.

**Questions For Authors:**

1) How is your method different from AD-GCL[1]? AD-GCL uses an adversarial method to enhance contrastive learning of graphs. It also uses a learnable method to drop edges which is quite similar as your method. Please clearly explain the key differences in the design of the two methods.
2) There are two main components proposed by PiNGDA, however the visualization part only shows the edge noise, how does the node attribute noise look like? The authors should provide more detailed explanation on the part.
[1]Suresh, S., Li, P., Hao, C., and Neville, J. Adversarial graph augmentation to improve graph contrastive learning, 2021

**Relation To Broader Scientific Literature:**

The paper proposes method about graph contrastive learning. Prior studies, such as DGI and GraphCL have demonstrated the effectiveness of contrastive objectives in learning graph representations, mainly using augmentations such as node/edge dropping. Newer methods such as GCA have introduced adaptable augmentation strategies. This paper extends these ideas by incorporating learnable noise into the contrastive framework, thereby enhancing representation robustness while maintaining theoretical consistency.

**Theoretical Claims:**

In Section 3 of this paper, the key theoretical assertions are meticulously validated through rigorous mathematical proofs. The authors clearly articulate their underlying assumptions, derive conclusions logically, and maintain coherence in the entire derivation sequence.

---

> ### Author Rebuttal · Authors · 2025-03-31
>
> **Response to Reviewer PRCX**
>
> We greatly thank you for the detailed and valuable comments. Please find our responses to the comments as follows:
>
> >**W1:** The depth of the discussion of the experimental results can be further improved, such as the explanation of certain experimental phenomena and the possible differences between different datasets. For example, in the Table 2, the learnable methods seem not to perform as well as the other two datasets.
>
> **Reply:** Regarding the observation that learnable augmentation performs worse than random augmentation on the WikiCS dataset, we believe this may be due to the following reasons.  Compared to other datasets, WikiCS has a relatively **higher number of average edges**. The learnable augmentation method may not effectively enhance model performance when modifying the edge structure. Moreover, the **node features** in WikiCS have relatively **low dimensionality**. It indicates that edges may be more important than features in WikiCS. As a result, modifying features without adjusting edges might lead to suboptimal performance. However, if the learnable edge augmentation cooperates learnable feature augmentation, it may yield better results as can be found in the last column of **Table 2**. We will include a more detailed analysis in the paper to further investigate the differences across datasets. Thank you again for your insightful feedback!
>
> >**W2:** The mathematical derivation in the method section can be more detailed to enhance readability and understanding. For example, the random variable alpha is a bit confusing. Why is this variable necessary? And why do you assume the probability distribution showed in equation 4? It will be very helpful if they are explained clearly.
>
> **Reply:** Thank you for your comments. In our paper, the random variable $\alpha$ is introduced to indirectly **measure the difficulty of the task**, closely tied to the concept of **task entropy**. We adjust the variance of the auxiliary Gaussian distribution, which directly influences the calculation of task entropy.
>
> >**Q1:** How is your method different from AD-GCL[1]? AD-GCL uses an adversarial method to enhance contrastive learning of graphs. It also uses a learnable method to drop edges which is quite similar as your method. Please clearly explain the key differences in the design of the two methods.
>
> **Reply:** Our method differs from AD-GCL primarily in its augmentation strategy and focus. From the aspect of motivation,  AD-GCL applies **adversarial graph augmentation** to minimize information from the original data which tries to maximize the loss, our approach enables the model to learn the augmentation simplify the representation learning.  From the aspect of augmentation, AD-GCL mainly targets **graph structure**, whereas our method considers **both graph structure and node attributes**, leading to a more comprehensive augmentation. This advantage is reflected in our superior results in node classification, as shown in **Table 1**.
>
> >**Q2:** There are two main components proposed by PiNGDA, however the visualization part only shows the edge noise, how does the node attribute noise look like? The authors should provide more detailed explanation on the part.
>
> **Reply:** Thank you for your insightful suggestion. From a contrastive learning perspective, methods like SimCLR and BYOL commonly use **Gaussian noise** as an augmentation, making it a natural choice for adaptation to **non-vision data (tabular data)**. This is also an advantage of our method which serves as a **unified framework** for both **graph** and **tabular node features**. Regarding visualization, node attributes are typically **high-dimensional**, making them difficult to represent intuitively. Even if converted into a matrix or heatmap, unlike in vision tasks, there is **no inherent spatial relationship between adjacent values**, which could lead to misleading interpretations. We appreciate your valuable feedback and will add further explanations to clarify this point.

---

### Decision · Program_Chairs · 2025-05-01

**Decision:**

Accept (poster)

**Comment:**

Thank you for submitting your work to ICML 2025, and for your efforts in the rebuttal phase to clarify the reviewers' concerns.
However, the current version fails to provide a theoretical and experimental comparison with some methods. The lack of detailed experimental settings may also hinder reproducibility. Additionally, the authors should include a deeper analysis of the experimental results. After reading the reviewers' comments and the authors' responses, I believe the authors can effectively address these issues in the camera-ready version. Since all the reviewers affirmed the novelty of the authors' work, I therefore recommend acceptance of the paper.